# Simultaneous nanocatalytic surface activation of pollutants and oxidants for highly efficient water decontamination

Ying-Jie Zhang[1,5], Gui-Xiang Huang [1,5], Lea R. Winter[2], Jie-Jie Chen[1], Lili Tian[3], Shu-Chuan Mei[1], Ze Zhang[4], Fei Chen[1], Zhi-Yan Guo[1], Rong Ji [3], Ye-Zi You [4], Wen-Wei Li [1], Xian-Wei Liu [1], Han-Qing Yu [1✉] & Menachem Elimelech [2✉]

Removal of organic micropollutants from water through advanced oxidation processes (AOPs) is hampered by the excessive input of energy and/or chemicals as well as the large amounts of residuals resulting from incomplete mineralization. Herein, we report a new water purification paradigm, the direct oxidative transfer process (DOTP), which enables complete, highly efficient decontamination at very low dosage of oxidants. DOTP differs fundamentally from AOPs and adsorption in its pollutant removal behavior and mechanisms. In DOTP, the nanocatalyst can interact with persulfate to activate the pollutants by lowering their reductive potential energy, which triggers a non-decomposing oxidative transfer of pollutants from the bulk solution to the nanocatalyst surface. By leveraging the activation, stabilization, and accumulation functions of the heterogeneous catalyst, the DOTP can occur spontaneously on the nanocatalyst surface to enable complete removal of pollutants. The process is found to occur for diverse pollutants, oxidants, and nanocatalysts, including various low-cost catalysts. Significantly, DOTP requires no external energy input, has low oxidant consumption, produces no residual byproducts, and performs robustly in real environmental matrices. These favorable features render DOTP an extremely promising nanotechnology platform for water purification.

[1] Department of Environmental Science and Engineering, University of Science and Technology of China, 230026 Hefei, China. [2] Department of Chemical and Environmental Engineering, Yale University, New Haven, CT 06520, USA. [3] School of the Environment, Nanjing University, 210023 Nanjing, China. [4] Department of Polymer Science and Engineering, University of Science and Technology of China, 230026 Hefei, China. [5] These authors contributed equally: Ying-Jie Zhang, Gui-Xiang Huang. ✉email: hqyu@ustc.edu.cn; menachem.elimelech@yale.edu

The intensification of industry- and consumer-derived organic micropollutants—such as pharmaceuticals, personal care products, and pesticides—poses significant challenges to maintaining ecological security and a clean water supply[1–5]. Effective, affordable, and environmentally benign treatment technologies for eliminating these micropollutants are thus highly desired[6–8]. Promising approaches include heterogeneous advanced oxidation processes (AOPs), where reactive species (e.g., reactive oxygen species (ROS)) are generated to degrade and mineralize organic pollutants, and the heterogeneous catalysts can be recycled to minimize environmental impacts[9–11].

Extensive research on heterogeneous AOPs for micropollutant removal from water has been conducted using various nanocatalysts (e.g., metallic oxides and carbonaceous nanomaterials) and persulfates (peroxymonosulfate (PMS) and peroxydisulfate (PDS))[12–15]. However, two fundamental questions remain for these persulfate-based heterogeneous catalytic systems. The first is the controversy regarding the catalytic mechanisms. Currently, four mechanisms of pollutant degradation and mineralization have been proposed, including sulfate/hydroxyl radicals, singlet oxygen, high-valent metals, and catalyst-mediated electron transfer[12,16–18]. However, solid evidence to support these mechanisms is still lacking. Another widely-overlooked inconsistency is the electron equivalent non-conservation contradiction in the reported PMS-based heterogeneous catalytic systems. Specifically, the electron equivalent consumed by the oxidant (calculated according to the actual dosage of PMS or PDS) is much lower than the donated electron equivalent from the pollutant (calculated according to the mineralized organics) (Table S1). This contradiction cannot be explained by the existing oxidation theory for pollutant degradation.

To answer these two fundamental questions, here we reinvented heterogeneous catalytic oxidation process for micropollutant removal. In contrast to the four AOP mechanisms for pollutant degradation and mineralization discussed above, we revealed that organic pollutant removal using a variety of nanocatalysts proceeds via a non-decomposing direct oxidative transfer process (DOTP). Such a process has the distinct advantages of rapid and complete pollutant removal, avoidance of toxic byproduct formation, and low oxidant consumption compared to the existing AOPs. The DOTP is also fundamentally different from adsorption, in which the transport of pollutants from the liquid bulk to a solid surface does not involve a change in molecular structure.

In this work, we present a systematic, in-depth investigation of persulfate-based heterogeneous catalytic oxidation processes over nanosized metallic oxides or carbonaceous materials, revealing new insights into the catalytic behavior and underlying mechanisms. We characterized the products and explored the reaction pathways, established the quantitative structure-activity relationship between the reaction pathways and the molecular structure of pollutants, and verified the elementary reaction steps of the surface reaction. We further revealed the origin of the catalytic activity in the persulfate-based heterogeneous decontamination systems, and we elucidated the activation, stabilization, and accumulation functions of the nanocatalyst surface underpinning the surface-dependent and thermodynamically-spontaneous DOTP. This new recognition of the DOTP resolves the controversial mechanisms and the electron-equivalent non-conservation issues in persulfate-based heterogeneous AOPs. Lastly, we evaluated the practical application potential of DOTP based on its catalytic performance, cost analysis, and applicability under environmentally relevant conditions. Overall, our work demonstrates high decontamination activity, long-term stability, robustness in real environmental matrices, and high potential of the DOTP for low-cost water purification applications.

## Results and discussion

**Revealing the heterogeneous catalytic direct oxidative transfer process.** $Co_3O_4$ has been commonly used as a model heterogeneous catalyst for persulfate-based AOPs involving the generation of radicals[15,19–21]. To reveal the heterogeneous catalytic oxidation mechanisms, we used nanosized $Co_3O_4$ and phenol (PhOH) as the model catalyst and pollutant, respectively, and employed PMS as the oxidizing agent (Figs. S1 and S2). The PhOH was completely removed from the system within 30 min, with almost full elimination of total organic carbon (TOC) and chemical oxygen demand (COD) in the bulk solution synchronously (Fig. 1a, b). However, the synchronous and efficient TOC removal could not be realized by degradation and mineralization in Fenton systems, which are typical AOPs, at the same dosage of oxidizing agent (2:1 dosage of oxidant to pollutant) (Fig. S3). These results indicated that the pollutant removal process observed in the $Co_3O_4$/PMS/PhOH system was not an AOP, but rather represented a new pollutant removal mechanism.

After the reaction in the $Co_3O_4$/PMS/PhOH system, 97% COD depletion was observed for the bulk solution, but the total reaction suspension (including both the solution and the suspended catalyst) exhibited only 18% COD removal (Fig. 1b) while the catalyst contained 77% of the original COD (Fig. 1c). These results indicated that ~80% of the COD was not degraded/mineralized, but rather was transferred from the bulk solution to the catalyst surface. Notably, such a COD transfer was unlikely to be due to simple adsorption, because the PMS-free control system ($Co_3O_4$/PhOH) exhibited no COD reduction in the solution (i.e., almost no adsorption occurred) (Fig. 1c). We conducted [14]C-PhOH isotope labeling to clarify the fate of the ~20% removed COD and found that all the labeled carbon remained in the suspension (containing solution and catalyst) after the reaction (without mineralization) (Fig. 1d). Therefore, in the removal of PhOH from the solution and transfer onto the catalyst surface (Fig. 1a, d), the TOC did not change, but the COD decreased by ~20% (Fig. 1b, c), suggesting that PhOH was oxidized in the transfer process. The oxidative transfer of PhOH from the solution to the catalyst surface was further verified by the energy dispersive spectra (EDS) of the reacted catalyst, which showed the accumulation of large amounts of carbon and oxygen with a C/O molar ratio of 6.33—approaching that of the PhOH molecule—after the reaction (Fig. 1e, f and Fig. S4). This result also suggests that the C and O skeletal structure of the PhOH derivatives that accumulated on the catalyst surface was preserved, indicating non-decomposing removal of PhOH. Overall, these results suggested a new class of pollutant removal process involving non-decomposing oxidative transfer to the catalyst surface—namely, a direct oxidative transfer process (DOTP).

Notably, the non-decomposing DOTP was found to be widely applicable beyond the $Co_3O_4$ catalytic oxidation system. We evaluated several other heterogeneous catalytic oxidation systems with different nanocatalysts, including FeMnO, $CuO_X$, CNT, and biochar (Supplementary Note 1); different organic micropollutants including industrial chemicals (phenol, aniline, and bisphenol A); pesticides (4-chlorophenol); preservatives (methylparaben); and pharmaceuticals and their intermediates (sulfanilamide, acetaminophens, and guaiacol); and different oxidizing agents including PMS and PDS. Through monitoring the TOC or COD changes, we determined the percentage contribution of DOTP to the total pollutant removal processes (DOTP ratios). The results showed that the pollutant removal in all these persulfate-based heterogeneous oxidative systems were dominated by DOTPs rather than AOPs (Fig. 1g, h and Figs. S5–S12).

**Identification of the reaction products and pathways in DOTPs.** We elucidated the reaction products and pathways of the

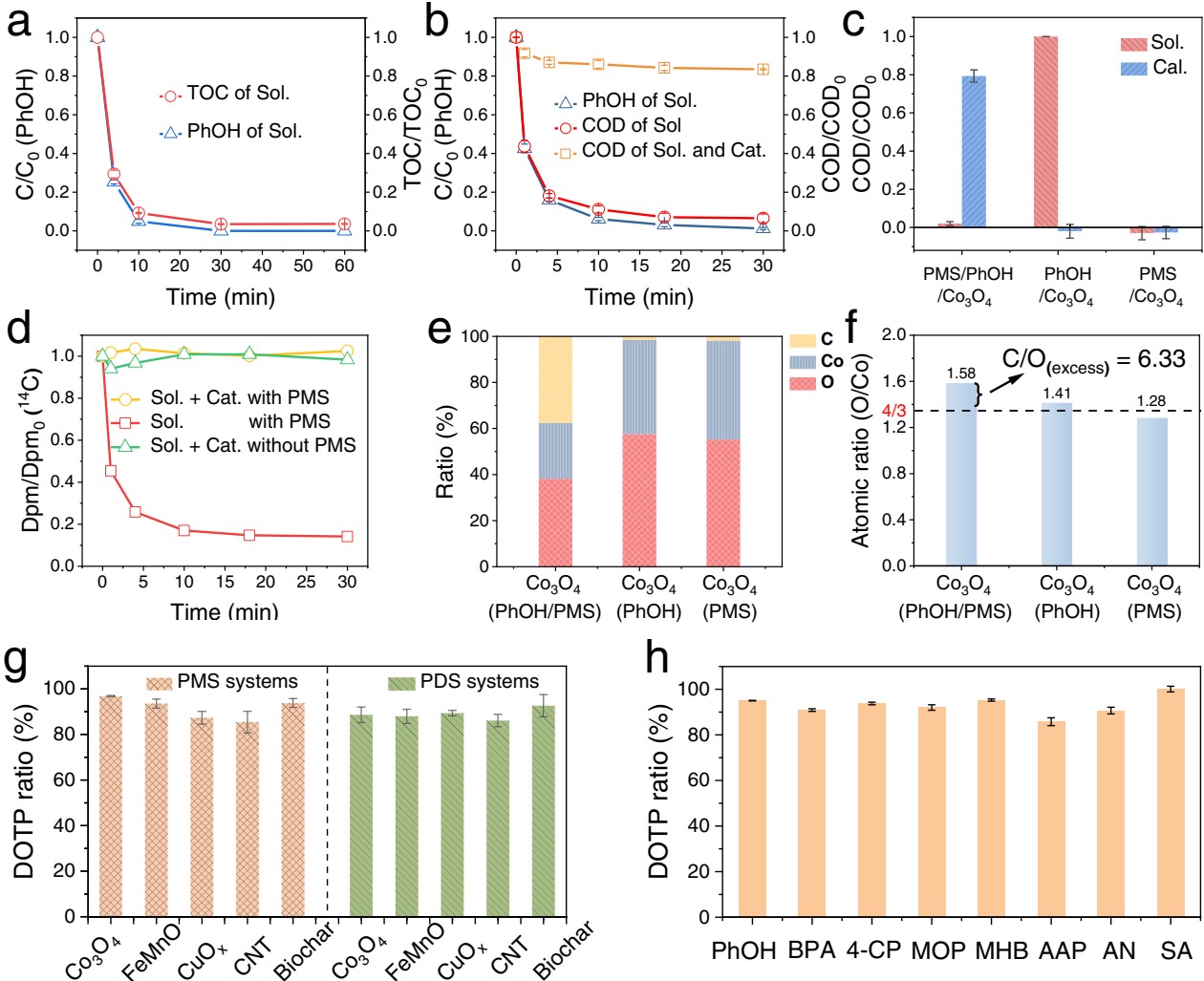

**Fig. 1 Removal of organic pollutant and DOTP ratios in the persulfate-based heterogeneous catalytic oxidation systems. a** Residual PhOH concentration and corresponding TOC removal in the $Co_3O_4$/PMS/PhOH system at PhOH dosage of 25 mg L$^{-1}$ (low-concentration system). **b** Residual PhOH concentration and corresponding COD removal in the $Co_3O_4$/PMS/PhOH system at PhOH dosage of 250 mg L$^{-1}$ (high-concentration system). **c** Residual COD of the solution and of the catalyst after reaction in the control systems of $Co_3O_4$/PMS/PhOH, $Co_3O_4$/PhOH, and $Co_3O_4$/PMS. **d** Changes in the radioactivity in the $^{14}$C-PhOH/$Co_3O_4$ systems with/without PMS. **e** EDS analyses of the elemental ratios in the catalyst after reaction in **c**. **f** Corresponding O/Co atomic ratios of the $Co_3O_4$ in **e**. **g** DOTP ratios for the removal of PhOH in the systems with different persulfates and nanocatalysts. **h** DOTP ratios for the removal of different organic pollutants in the $Co_3O_4$/PMS and CNT/PDS systems. All the dosages of oxidants (PMS and PDS) used above (**a**–**h**) were twice the molar amount of pollutants. Sol. solution, Cat. catalyst, Dpm. disintegration per minute, C/O$_{excess}$ the atomic ratio of increased C and O on the $Co_3O_4$ after the reaction, 2,6-M-PhOH 2,6-dimethylphenol, BPA bisphenol A, 4-CP 4-chlorophenol, MOP guaiacol, MHB methylparaben, AAP acetaminophen, AN aniline, SA sulfanilamide, DOTP ratio the proportion of pollutant removal accomplished via the DOTP reaction process.

DOTP in the representative $Co_3O_4$/PMS/PhOH system. Significant accumulation of C-O bond-containing organic products on the catalyst surface was confirmed by multiple characterizations, including transmission electron microscopy (TEM), scanning transmission electron microscopy-energy dispersive spectroscopy (STEM-EDS) mapping, X-ray photoelectron spectroscopy (XPS), thermal gravimetric analysis-Fourier transform infrared spectroscopy (TGA-FTIR), and Fourier transform infrared spectroscopy (FTIR). The light contrast in the TEM images on the reacted $Co_3O_4$ (Fig. 2a and Figs. S13–S15) and the narrow decomposition temperature range in air (300 ± 20 °C) (Fig. 2b, c) were characteristic of polymeric species rather than small molecules bound to the catalyst via physical adsorption[22,23]. Further, the XPS and FTIR spectra showed the presence of C-O-C, C-C, and C-O bonds, suggesting the probable formation of polyphenylene oxides (PPO, also known as polyphenyl ethers) on the $Co_3O_4$ surface

(Fig. 2d, e and Figs. S16 and S17). The quantitative analysis of carbon mass loss during thermal decomposition also verified a complete transfer (~100%) of the pollutants onto the catalyst surface (Fig. 2b and Fig. S18).

To further identify the structures of the products formed on the $Co_3O_4$ surface, we attempted to separate the deposited substances from the catalyst using various aqueous solutions and organic solvents. However, acidic and alkaline aqueous solutions, ethanol, chloroform, and toluene were not effective in eluting the major accumulated organics (Fig. 3a and Fig. S19). Such high resistance to desorption suggests that the majority of the accumulated substances might be formed from a crosslinking polymerization of PhOH on the catalyst surface. The crosslinking reaction of PhOH can be triggered readily on the three active hydrogen atoms of the benzene ring (i.e., the *ortho*- and *para*-positions of the hydroxyl group), and such reactions are common

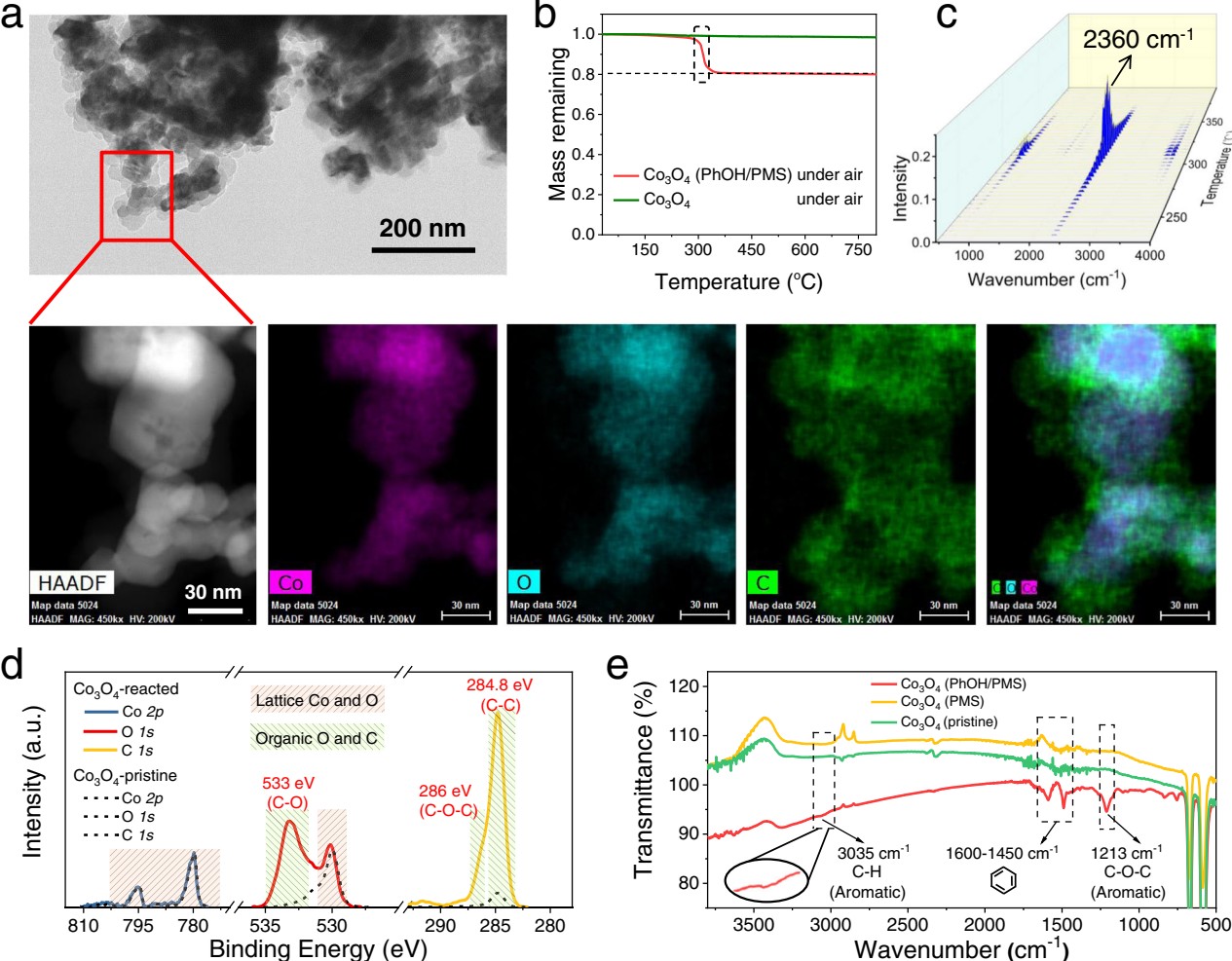

**Fig. 2 Product analyses of the model reaction system (Co₃O₄/PMS/PhOH). a** STEM, HAADF, and EDS elemental mapping images of the Co₃O₄ after the reaction. **b** TGA curves of the pristine and reacted Co₃O₄ in air (O₂). The mass loss of 20% for the reacted Co₃O₄ was equal to the initial concentration ratio of [PhOH] to [PhOH] + [Co₃O₄], which indicates that the pollutant molecules were fully transferred to the catalyst surface. **c** 3D-FTIR spectra of the gas products detected from TGA of the reacted Co₃O₄ in **b**. The decomposition temperature (centered Around 300 °C) in air (O₂) and the gas product (CO₂) are characteristic of polymers. **d**, **e** XPS spectra (**d**) and FTIR spectra (**e**) of the pristine and reacted Co₃O₄. The signal intensities in the XPS spectra of the pristine and reacted Co₃O₄ were normalized by that of Co 2p.

in the polymer engineering field to synthesize PPO[24,25]. Such crosslinking polymerization of PhOH may result in polymers that resist desorption. In contrast, the substitution of two of the three active hydrogen atoms (e.g., two *ortho*-positions) in the phenol molecule with methyl should result in the formation of a non-crosslinked state and facile desorption of the product from the catalyst (Fig. 3b).

To test this hypothesis and identify the pathways in DOTPs, we conducted hypothetico-deductive experiments by using 2,6-dimethylphenol (2,6-M-PhOH) as the pollutant, which should not form crosslinked products following the DOTP. As expected, 2,6-M-PhOH in the bulk solution was almost entirely transferred to the catalyst surface after the reaction (Fig. S20) and could be readily removed from the surface upon washing with solvents (i.e., non-crosslinked state). This in turn proved that the products that could not be eluted in the PhOH system were crosslinked. For the 2,6-M-PhOH system, ~85% of the products were dissolved by ethanol, and the remaining ~15% were dissolved by toluene (Fig. 3c and Fig. S21). We used liquid chromatography-mass spectrometry (LC-MS), gel permeation chromatography (GPC), matrix-assisted laser desorption/ionization time-of-flight mass spectrometry (MALDI-TOF-MS), and

nuclear magnetic resonance (NMR) spectroscopy to analyze the dissolved products. The substances dissolved in the ethanol solution were identified as biphenyl quinone compounds (i.e., 3,3′,5,5′-tetramethyldiphenoquinone) (Fig. 3d, e). These compounds could be generated by the coupling reaction (C-C coupling) of 2,6-M-PhOH (CR pathway) (Fig. 3f)[26,27]. The other substances dissolved in toluene were identified as chain-like polyphenyl ethers with a molecular weight of ~6062 Da (Fig. 3g, h and Fig. S22). These compounds could be generated by the polymerization reaction (C-O polymerization) of 2,6-M-PhOH (PR pathway) (Fig. 3i)[28,29]. Overall, the hypothetico-deductive results demonstrated that the crosslinked products in the PhOH system were mainly PPO, and that coupling and polymerization reaction pathways are responsible for the PhOH and 2,6-M-PhOH oxidative transfers from the bulk solution to the Co₃O₄ surface.

**Quantitative structure–activity relationship analysis of DOTP reactions with phenols.** The proportions of pollutant removal contributed by the coupling and polymerization pathways differed significantly for PhOH and 2,6-M-PhOH. To understand

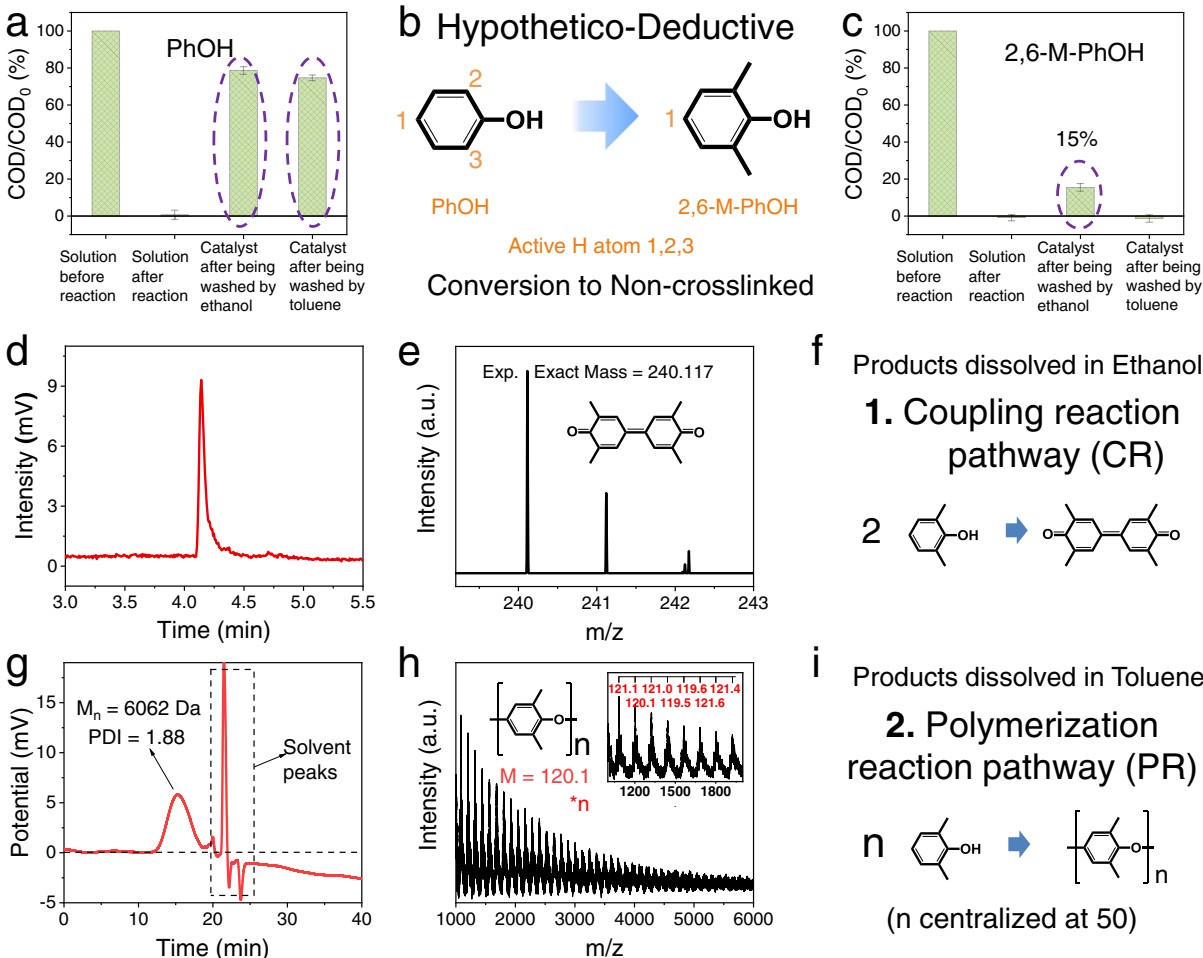

**Fig. 3 Reaction pathway analyses of DOTP. a** Residual COD of the reaction solution and of the reacted $Co_3O_4$ after elution with different solvents in the $Co_3O_4$/PMS/PhOH system. **b** Active hydrogen atoms of PhOH and 2,6-dimethylphenol. **c** Residual COD of the reaction solution and of the reacted $Co_3O_4$ after elution with different solvents in the $Co_3O_4$/PMS/2,6-dimethylphenol system. **d**, **e** Liquid chromatogram (**d**) and mass spectrum (**e**) of the products washed off by ethanol in **c**. The theoretical exact masses of 3,3,5,5-tetramethyl-4,4'- biphenylquinone including isotopes are 240.12 (100%), 241.12 (17.3%), 242.12 (1.4%), consistent with the experimental values in **e**. **f** Schematic of the coupling reaction pathway. **g**, **h** Gel permeation chromatogram (**g**) and MALDI-TOF-MS spectrum (**h**) of the products washed off by toluene in **c**. The polymer dispersity index (PDI) of 1.88 indicates that the molecules centrally weighted at 6062 Da had a narrow weight distribution. The inset in **h** is the partially enlarged view of the MALDI-TOF-MS spectrum, in which the mean mass interval of 120.1 matches the polymeric unit of poly (2,6-dimethyl-1,4-phenylene oxide) (PPO). **i** Schematic of the polymerization reaction pathway. Exp. experimental, $M_n$, number-average molecular weight, $M$ theoretical value of the molar mass of the polymeric unit, $N$ degree of polymerization.

how the reaction pathways in DOTP are affected by pollutant molecular structure, we analyzed the quantitative structure/ activity relationship (QSAR) of several pollutants, including PhOH, 2,6-M-PhOH, bisphenol A (BPA), and 4-chlorophenol (4-CP). Both the coupling and polymerization pathways contributed to the oxidative transfer process for all these compounds, but with different proportions depending on the number of active hydrogen atoms (on the *para-* and *ortho*-positions of the functional group) on the benzene ring of the pollutant molecules (Table 1). Meanwhile, the solubility of the products and the amount of oxidant consumed also varied with the number of active hydrogen atoms (Figs. S23 and S24).

For the pollutant molecules with one to two active hydrogen atoms, such as 2,6-M-PhOH and 4-CP, the C-C coupling pathway was dominant in DOTP, and a small amount of polyphenyl ethers with a chain-like structure were formed through a minor C-O polymerization pathway (Table 1). After the DOTP reaction, all the products on the catalyst surface could be dissolved completely (Fig. S23). In comparison with the

conventional Fenton systems that typically require an $H_2O_2$ dosage that is 100 times larger than the pollutant amount[30–32], DOTPs showed extremely low PMS consumption (approximately one PMS molecule consumed for each pollutant molecule removed) (Fig. S24b, d). For the pollutant molecules with three to four active hydrogen atoms, such as PhOH and BPA, polymerization became the dominant pathway, in which poly-phenyl ethers with a crosslinked structure were generated and slightly more PMS was consumed (approximately 2 PMS molecules consumed for each pollutant molecule removed) (Table 1 and Fig. S24a, c). These crosslinked products could not be dissolved from the catalyst surface (Fig. 2a and Figs. S13–S15 and S23a).

Based on the QSAR analysis, the reaction rules for different pollutants in DOTPs were deduced. We found that the reaction pathways (CR and PR) for different pollutants were highly dependent on their molecular structures, especially the number of active H atoms. Although a polymerization of pollutants on certain catalysts has been noted in several previous studies[33–35],

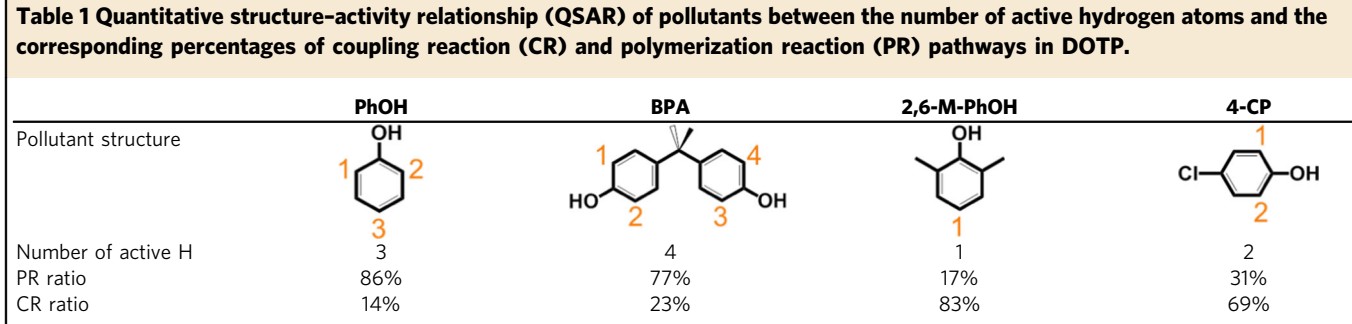

**Table 1 Quantitative structure–activity relationship (QSAR) of pollutants between the number of active hydrogen atoms and the corresponding percentages of coupling reaction (CR) and polymerization reaction (PR) pathways in DOTP.**

|  | PhOH | BPA | 2,6-M-PhOH | 4-CP |
|---|---|---|---|---|
| Pollutant structure | | | | |
| Number of active H | 3 | 4 | 1 | 2 |
| PR ratio | 86% | 77% | 17% | 31% |
| CR ratio | 14% | 23% | 83% | 69% |

this phenomenon was only considered as a selective oxidation in specific systems. In addition, the coupling reaction has not yet been reported, and the pollutant removal process and reaction mechanism in the persulfate-based heterogeneous catalytic field are still controversial as AOPs. Here, we qualitatively and quantitatively revealed the PR and CR pathways, and we also clarified for the first time that the pollutant removal process in the persulfate-based heterogeneous catalytic oxidation systems is dominated by DOTPs rather than AOPs. In the following discussion, we further reveal the pollutant removal PR and CR mechanisms.

**Reaction mechanism and elementary reaction steps of DOTP.** To shed light on the reaction mechanism of DOTP, we investigated the reaction stoichiometry and kinetics of PhOH removal under various dosage ratios of PMS to PhOH (Fig. 4a). We found that the stoichiometric ratio between the consumed amounts of PMS and PhOH was ~1.8 in the $Co_3O_4$ catalytic system (Fig. 4b). At a PMS to PhOH dosage ratio of 1.8:1, the reaction followed second-order kinetics (Fig. 4c) rather than pseudo-first-order kinetics (Fig. S25); the latter is typical for conventional ROS-based AOPs[17–19,36]. Similar results were obtained when other catalysts such as metal oxides (FeMnO) and carbonaceous materials (biochar) were used (Figs. S26–S28). These results imply that in these heterogeneous catalytic systems, a direct redox reaction (i.e., bimolecular reaction) between PhOH and PMS might occur[37,38].

The direct redox reaction mechanism and the importance of the catalyst surface were verified by galvanic cell experiments (Fig. 4d). In separated galvanic cells (i.e., two-chamber system), PMS (i.e., oxidant) and PhOH (i.e., reductant) underwent the redox reaction (i.e., electron transfer) through the circuit, and the reaction (removal of PhOH) could only occur on the catalyst surface (Fig. 4e and Fig. S29). Compared with the single-chamber reaction system (PMS and PhOH contact directly), the reaction rate of the two-chamber system substantially decreased due to the separation of PMS and PhOH at two electrodes. Similar results were obtained using an analogous redox reaction system involving direct electron transfer between PMS and KI (Fig. 4f). The requirement of simultaneous contact between the oxidant and pollutant with the same catalyst surface, as well as the similar effect of separation on the PMS/PhOH system as on the PMS/KI system, supported the occurrence of a direct redox reaction mechanism in DOTPs.

In addition, since the reaction between PMS and KI is a $2e^-$ redox process (PMS was reduced to $SO_4^{2-}$ without generation of $SO_4^{\bullet-}$ or $\bullet OH$) (Fig. S30)[39–41], the analogy experiments suggested that the PMS-PhOH reaction also involved a $2e^-$ direct electron transfer process. This was verified by in-situ EPR experiments (Supplementary Note 2) (Fig. 4g, h and

Figs. S31–S33), radical quantitation fluorescence detection (Fig. 4i and Fig. S34), and scavenging experiments (Fig. S35), which showed that no radicals were involved in the PMS-PhOH reaction. Taken together, the reaction stoichiometry and second-order kinetics as well as the results of galvanic cell experiments, analogy experiments (PhOH/KI), in-situ EPR detection experiments, and radical quantitation and scavenging experiments suggested that the reaction between PMS and PhOH is a $2e^-$ direct redox reaction catalyzed by a heterogeneous catalyst surface. This mechanism is distinct from the catalyst-mediated electron transfer mechanisms reported for AOPs, which have been described as two types: (1) the pollutant first transfers electrons to the catalyst, and then the catalyst transfers electrons to PMS/PDS, or (2) the PMS/PDS first forms an activated complex with the catalyst, and the complex then oxidatively degrades the pollutant[17,42].

Following the direct redox reaction mechanism, we further explored the elementary steps of the DOTP on the heterogeneous catalyst surface. As shown in Fig. 5a, a two-electron transfer step first occurs between PhOH and PMS, similar to the common oxidative polymerization reaction of PhOH catalyzed by a copper-ammonia complex and with $O_2$ as the oxidant in the polymer synthesis field[28,43] (Supplementary Note 3). This direct electron transfer would result in the formation of phenoxonium ions, which might be stabilized on the catalyst surface and then undergo instantaneous resonance transfer to form a positively charged center on the carbon at the *ortho-* and *para-* positions of phenolic hydroxyl[44,45]. This resonance transfer would induce the stabilized phenoxonium ions either to spontaneously undergo C-O polymerization (PR) with phenols or to trigger C-C coupling (CR) with another intermediate at the positively charged center of the C atom. Since the electrophilic phenoxonium ions can be scavenged by stronger nucleophilic reagents such as acetonitrile, the reaction of PhOH should be inhibited by acetonitrile. This effect was explored using experiments with acetonitrile addition (Fig. 5b and Fig. S36), which resulted in nucleophilic inhibition and provided evidence supporting the generation of phenoxonium ion intermediates[46]. In addition, reaction systems using $D_2O$ as the solvent exhibited a secondary kinetic isotope effect (KIE) (Fig. 5c, d and Figs. S37–S38), indicating that the C atoms attached to the isotope atoms underwent a transition from $sp^3$ hybridization to $sp^2$ hybridization[47]. The rearrangement of H in the polymerization pathway and the departure of H in the coupling pathway were controlled by the solvent and were the source of the transition from $sp^3$-C to $sp^2$-C (Fig. 5a). Therefore, the nucleophilic inhibition and KIE experiments further verified the proposed reaction mechanism and elementary reaction steps.

**Origin of the heterogeneous catalytic oxidation activity for DOTPs.** DOTP molecular reactions described above (Fig. 5a)

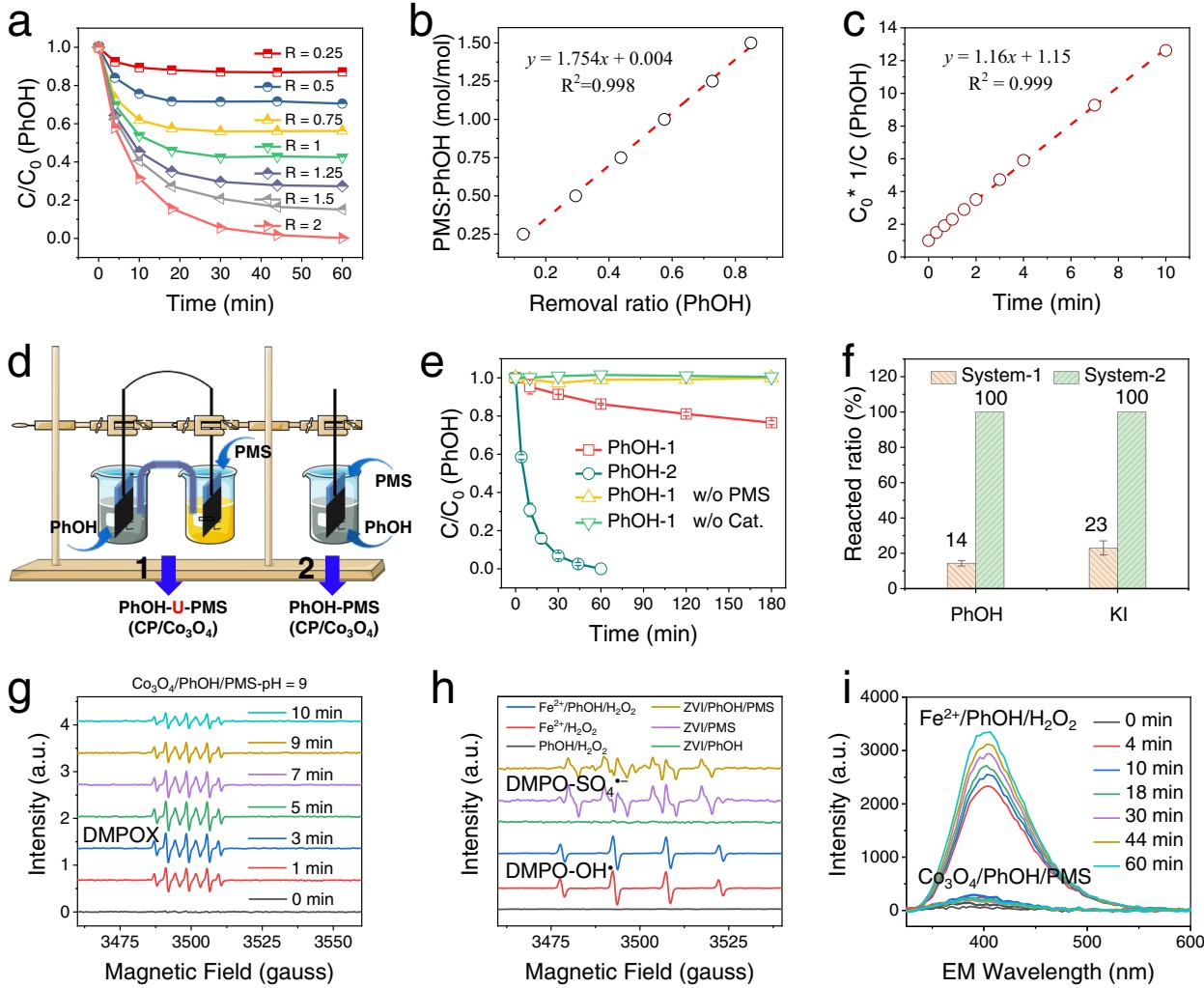

**Fig. 4 Reaction stoichiometry, kinetics, and direct oxidation mechanism of DOTP. a** Concentration profiles of PhOH at different dosage ratios of PMS to PhOH. **b** Regression curve of the dosage ratio of PMS to PhOH versus the removal ratio of PhOH. **c** Second-order kinetics fitting of the reaction with PMS to PhOH dosage ratio of 1.8. **d** Schematic illustration of the two-chamber galvanic cell system (marked as 1) and the single-chamber contact-type system (marked as 2). **e** Concentration profiles of PhOH removal in the two systems in **d**. **f** Conversion of the reactants in the analogy experiments between PhOH and KI in the two systems in **d**. The reactions of PhOH in the two systems exhibited the same trend as those of KI within 60 min. **g** Time-dependent EPR spectra of the $Co_3O_4$/PMS/PhOH system with DMPO as the spin trapping agent. **h** EPR spectra of the $Fe^{2+}/H_2O_2$/PhOH (Fenton) and ZVI/PMS/PhOH systems recorded at a reaction time of 5 min with DMPO as the spin-trapping agent. **i** Fluorescence emission spectra of the $Fe^{2+}/H_2O_2$/PhOH (Fenton) and $Co_3O_4$/PMS/PhOH systems for quantitative detection of radicals with benzoic acid (BA) as the probe molecule (the excitation wavelength was 310 nm). Differing from the typical radical systems (i.e., •OH in the $Fe^{2+}/H_2O_2$/PhOH system and $SO_4^{•-}$ in the ZVI/PMS/PhOH system) that exhibited DMPO-OH• (with hyperfine couplings $a_N = a_{\beta-H} = 14.9$ G) or DMPO-$SO_4^{•-}$ (with hyperfine splitting constants of $a_N = 13.2$ G, $a_{\beta-H} = 9.6$ G, $a_{\gamma-H1} = 1.48$ G and $a_{\gamma-H2} = 0.78$ G)) signals in the EPR spectra, in the $Co_3O_4$/PMS/PhOH system, only the DMPOX signal (with a 1:1:1 triplet ($a_N = 7.2$ G) of 1:2:1 triplet ($a_{\gamma-H} = 4.1$ G, 2H)) appeared from the beginning of the reaction and then exhibited initial increasing and later decreasing. The result of the in situ EPR experiments indicates no •OH and $SO_4^{•-}$ formation in the $Co_3O_4$/PMS/PhOH system. $\text{[structure]}$ + PMS (HO-O-$SO_3^-$) $\xrightarrow{\text{nanocatalyst surface}}$ $\text{[structure]}$ + $SO_4^{•-}$ + H$^+$R, molar ratio of PMS to PhOH; CP, carbon paper; Reacted ratio, (1-C/$C_0$)*100%; ZVI, zero-valent iron solid; DMPO, $\text{[structure]}$ ; DMPO-X, $\text{[structure]}$.

were simulated by DFT quantum chemical calculations to reveal the origin of heterogeneous catalytic activity. As shown in Fig. 5e, Figs. S39–S41, and Table S2, the thermodynamic free energies for the reactions catalyzed by the $Co_3O_4$ surface were energetically downhill, from the reactants (i.e., PhOH and PMS) to the phenyl-oxygen and phenyl-carbon intermediates and then to the corresponding polymerization and coupling products. Therefore, the overall catalytic reactions occurring on the catalyst surface were spontaneous and could occur at room temperature due to the low activation barriers (Fig. S42). However, once the stabilizing effect of the catalyst surface was removed for comparison, all the

reaction steps for the generation of the pollutant intermediates became non-spontaneous with uphill thermodynamic free energies and higher activation barriers (Fig. 5e and Figs. S43 and S44). These results suggest that the catalyst surface plays an essential role in DOTPs. Specifically, the low energy barrier of PMS/PhOH reaction on the catalyst surface, the low thermodynamic free energy of the stabilized pollutant intermediates, and the low thermodynamic free energy of the accumulated polymerization products and coupling products correspond respectively to three important functions of nanocatalyst surface, namely, activation, stabilization, and accumulation (Supplementary Note 4).

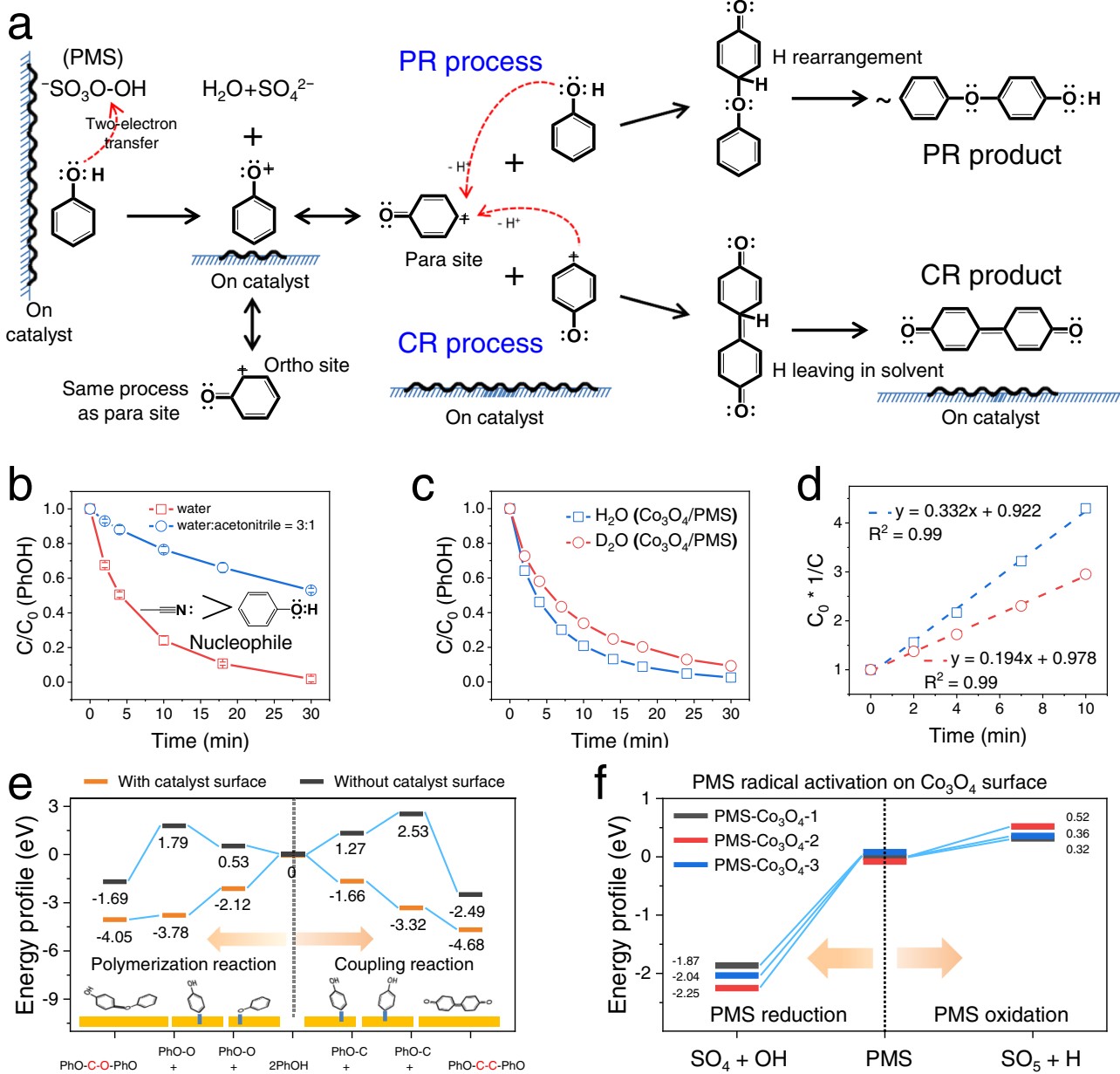

**Fig. 5 Evolution process of DOTP and thermodynamic feasibility analysis. a** Proposed elementary reaction pathways of the oxidative coupling and polymerization of PhOH on the catalyst surface in the DOTP. **b** Competitive inhibition of acetonitrile on PhOH removal. **c** Removal processes of PhOH in $H_2O$ and $D_2O$. **d** Corresponding kinetic curves of **c**. The KIE value was calculated to be $K_H/K_D$ ~1.4, which is in the range of secondary kinetic isotope effects. **e** Thermodynamic potential energy curves during the reaction in DOTP. **f** Thermodynamic potential energy curves of the reductive and oxidative activation processes of PMS on the $Co_3O_4$ surface in the AOP. PR polymerization reaction, CR coupling reaction.

$Co_3O_4$ has been accepted as a model catalyst in AOPs to activate PMS for free radicals (•OH or $SO_4^{•-}$) generation and pollutant degradation/mineralization[15,19]. In order to evaluate the extent to which the radical pathway may occur, we calculated the energetics of the AOP reactions with $Co_3O_4$/PMS for comparison. The results showed that the oxidative reaction pathway of PMS was non-spontaneous; thus, the formation of free radicals was unfavorable via PMS activation on the $Co_3O_4$ surface (Fig. 5f and Fig. S45). Since the reductive and oxidative reactions of PMS could not occur cyclically to produce free radicals continuously, an AOP was not favorable to occur in these persulfate-based heterogeneous catalytic oxidation systems. The thermodynamic simulation results strongly suggest that surface oxidative transfer of pollutants (CR and PR pathways) rather than degradation and

mineralization (via radicals) should be the predominant processes for pollutant removal from water in heterogeneous catalytic oxidative environments.

The above experimental and theoretical results show that DOTPs are fundamentally different from AOPs (Fig. 6). In conventional AOPs, efficient pollutant removal entails degradation and mineralization via ROS, which is generated by raising the potential energy of the oxidizer (e.g., $H_2O_2$ 1.78 eV, PMS 1.82 eV; their oxidative species: •OH 2.70 eV, $SO_4^{•-}$, 3.10 eV) at the cost of high energy and intensive chemical input[48]. In contrast, the DOTP also increases the potential difference between pollutants and oxidants, but in a fundamentally different way: the pollutants and oxidants are simultaneously activated by the catalyst surface (Fig. S40). The activated species then undergo

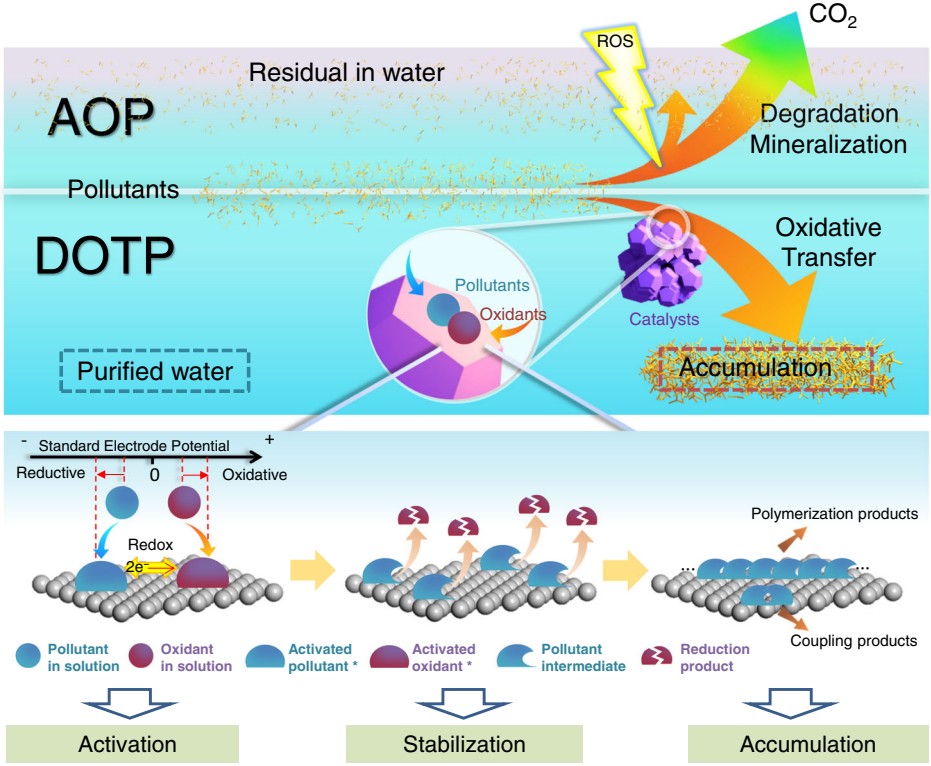

**Fig. 6 Schematic illustration of DOTP and AOP.** The surface (heterogeneous nanocatalysts) functions (i.e., activation, stabilization, and accumulation) of DOTP and the comparison with AOP for water purification.

spontaneous, direct redox reactions to form pollutant oxidation intermediates, which are stabilized by the catalyst surface. The stabilized intermediates subsequently accumulate on the heterogeneous catalyst surface via coupling and polymerization reactions. The three functions (i.e., activation, stabilization, and accumulation) of the nanocatalyst surface make the DOTP process thermodynamically spontaneous and kinetically easy to proceed.

**Evaluation of DOTP in practical applications**. DOTPs may be exploited for the development of new oxidative water purification technologies. Consistent with the thick layer state of aggregated products that formed on the catalyst surface (Fig. 2a and Figs. S13-S15), the DOTP showed a high removal capacity and long-term stability (at least 15 cycles) for removing pollutants at ~10 mg L$^{-1}$ initial concentration level (Figs. S46a and S47a). After the surface became saturated, the catalyst could be regenerated by thoroughly removing the accumulated organic matter from the catalyst surface via elution with organic solvents or an annealing treatment in air (Figs. S46b and S47b). The water purification performance of DOTP was compared with that of AOPs and adsorption processes. Two typical free radical systems (i.e., Fe$^{2+}$/H$_2$O$_2$ and ZVI/PMS) were adopted as representative AOPs (Fig. 7a–c). Compared with the AOPs, DOTP showed a significantly lower consumption of oxidizing agents as well as a much higher ratio of TOC removal (less residual or secondary contamination) (Supplementary Note 5).

When compared with adsorption (physical process) using activated carbon as the representative adsorbent (Figs. S48 and S49, Fig. 7d, e, and Table S3), the DOTP using activated carbon as the catalyst in a PDS oxidation system (AC/PDS) showed a relatively higher decontamination capacity and longer stability (Supplementary Note 6). In Fig. 7d, e, the DOTP showed near-complete removal of pollutants for more than 150 h, whereas the

adsorption process became ineffective for pollutant removal after only 24 h; this was further confirmed by the scale-up experiment results in Fig. S49. Furthermore, we estimated a cost comparison for these processes. The cost of a DOTP for removal of 1 kg PhOH ($11.6) was much lower than that of adsorption ($33.4) and AOPs (>$212.4) (Tables S4 and S5), indicating that DOTP is promising as the low-cost, sustainable water purification technology.

As oxidation technologies, DOTPs are more resistant than AOPs to environmental interferences, such as the presence of natural organic matter and chloride ions[49,50], because DOTPs does not involve any highly reactive species (e.g., radicals) (Fig. S50). Hence, we extended DOTP to the treatment of various micropollutants (i.e., bisphenol A, 4-chlorophenol, and sulfamethoxazole) at environmental concentrations (μg L$^{-1}$) by using a homemade membrane catalytic reactor (Fig. 7f). The results showed that the micropollutants could be effectively removed and that the high DOTP ratio (~98.2%) could be maintained under environmentally relevant conditions, such as low concentration and real water matrix (Fig. 7g, h and Figs. S51–S55). Therefore, DOTPs will be highly effective for the selective removal of micropollutants that exist at low environmental concentrations and are difficult to remove using current water treatment processes[4,51]. Because of the high decontamination capacity, long-term stability, low cost, and excellent resistance to interference from environmental impurities, DOTP technologies hold great potential for water purification.

**Outlook**. In this work, we revealed direct oxidative transfer processes (DOTPs) for aquatic pollutant removal via heterogeneous catalytic oxidation as processes distinct from conventional advanced oxidation and adsorption processes, and systematically elucidated the reaction mechanisms. We found that DOTPs can efficiently remove pollutants with electron-donating groups, such

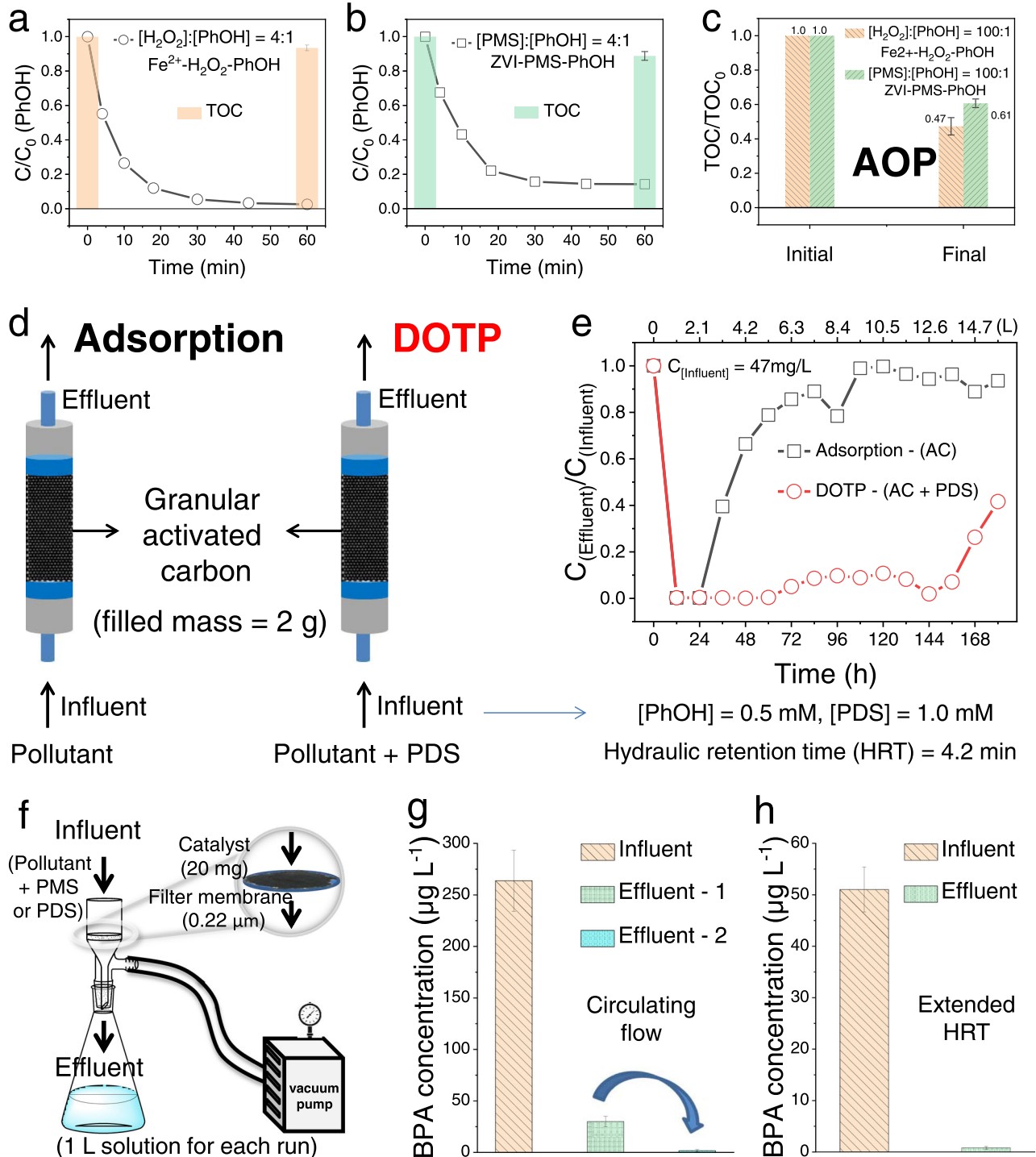

**Fig. 7 Evaluation of the practical application of DOTP, AOP, and adsorption processes. a, b** Pollutant removal and TOC elimination in the typical $Fe^{2+}$/ $H_2O_2$ (**a**) and ZVI/PMS (**b**) AOP systems, with oxidants to pollutants dosage ratios of 4:1. **c** Final removal of TOC in the two typical AOP systems at an oxidant dosage 100 times the molar concentration of the pollutant. **d, e** Water purification effects reflected by the continuous-flow operation of DOTP and adsorption illustrated in fixed bed reactor. **f** Schematic illustration of the membrane catalytic reactor (catalyst mass = 20 mg). **g, h** Detected BPA concentrations in the influent and effluent for 1-L tap water with an initial BPA concentration approximately at (**g**) 250 μg L$^{-1}$ and (**h**) 50 μg L$^{-1}$. For **g**, the first-run effluent (effluent-1) was reflowed over the membrane catalytic reactor for further BPA removal to obtain the second-run effluent (effluent-2). For **h**, the hydraulic retention time (HRT) was extended from 20 min/L (in **g**) to 40 min L$^{-1}$. The circulating flow of the first-run effluent in **g** and the extension of HRT in **h** were used for increasing the contact efficiencies between the pollutant/oxidant and the catalyst surface. These two methods were both effective to reduce the influence of diffusion control of micropollutants at low concentrations and thus increase their removal efficiency. ZVI zero-valent iron, AC activated carbon.

as phenols and anilines, with low oxidant dosages and no external energy input, which results in a favorable cost profile. Unlike adsorption, DOTPs do not require nanomaterials with a high specific surface area. Thus, many inexpensive nanocatalysts, such as metal oxides and carbonaceous materials, may be used to achieve excellent DOTP performance. For example, biochar obtained based on carbonaceous sludge precursors in sewage treatment plants may further reduce the application cost of DOTP. In addition, DOTP occurs in the heterogeneous catalytic oxidation environment (nanocatalysts and oxidizing agents), where the oxidizing agents merely serve as electron acceptors (i.e., oxidant) and do not directly participate in the surface coupling and polymerization reactions in the later stage of DOTP (from the oxidative pollutant intermediates to products). Thus, other oxidizing agents beyond persulfates may also be applicable for DOTP. Since DOTP has the ability to recover organic matter from water, it may potentially enable the reuse of organic contaminants as recoverable valuable resources.

## Methods

**Materials**. Nano cobalt (II, III) oxide ($Co_3O_4$, 30 nm, 99.5%), phenol (PhOH, 99.5%), bisphenol A (BPA, 99.8%), 2,6-dimethylphenol (2,6-M-PhOH, 99.5%), p-chlorophenol (4-CP, 99%), guaiacol (MOP, 98%), methylparaben (MHB, 99%), acetaminophen (AAP, 99.5%), potassium dichromate ($K_2Cr_2O_7$, 99.9%), potassium iodide (KI, 99.5%), zero-valent iron powder (ZVI, 98%), and powdered activated carbon (AC) were purchased from Aladdin Co., China. Peroxymonosulfate (PMS, $2KHSO_5 \cdot KHSO_4 \cdot K_2SO_4$, 4.5% active oxygen) was purchased from Beijing J&K Co., China. Potassium persulfate (PDS, 99.5%), hydrogen peroxide ($H_2O_2$, 30%), ascorbic acid (AA, 99.5%), concentrated $H_2SO_4$ (GR), boric acid, sodium tetraborate decahydrate, methanol, ethanol, toluene, chloroform, tetrahydrofuran (THF), sodium sulfate ($Na_2SO_4$), potassium hydroxide (KOH), hydrochloric acid (HCl), iron sulfate heptahydrate ($FeSO_4 \cdot 7H_2O$, 99%), and granular activated carbon (GAC, 12-20 mesh) were purchased from Shanghai Chemical Reagent Co., China. Silver sulfate ($Ag_2SO_4$) and mercuric sulfate ($HgSO_4$) were purchased from Shanghai Macklin Biochemical Co., China. Chloroform-d ($CDCl_3$-d, 99.8 atom % D), dimethyl sulphoxide-$d_6$ (DMSO-$d_6$, 99.8 atom % D), deuterium oxide ($D_2O$, 99.9 atom % D), Nafion solution (5 wt%), 5,5-dimethyl-1-pyrroline-N-oxide (DMPO), and acetonitrile were purchased from Sigma-Aldrich Co., China. Carbon papers were purchased from Toray Co., Japan. $^{14}C$-phenol was purchased from Cambridge Isotope Laboratories, USA. CNTs (95 wt%, 5-15 nm, 3-12 μm) were purchased from Shenzhen Sixthelement Co., China. FeMnO nanospheres and biochar were prepared according to previously reported protocols[18,52]. Specifically, for FeMnO, 25 mL aqueous solution of $MnCl_2 \cdot 4H_2O$ (6.25 mmol) and poly-vinylpyrrolidone (PVP, 0.75 g) was mixed with $K_3[Fe(CN)_6]$ (125 mM, 25 mL). After stirring for 30 min and ageing for 1 day, the collected precipitate was dried and calcined at 400 °C for 1 h in air. For biochar, the collected carbonaceous sludge from a local wastewater treatment plant was dried and carbonized at 600 °C for 4 h under $NH_3$/Ar (10/90, v/v) atmosphere. $CuO_x$ was prepared by calcining $Cu(CH_3COO)_2 \cdot H_2O$ in air at 400 °C for 2 h. Unless otherwise specified, all chemicals were used as received without further purification.

**Batch experiments**. For the batch experiments of pollutant removal, unless otherwise specified, the molar ratio of oxidants to pollutants was fixed at 2:1 as the initial reaction condition. For the $Co_3O_4$/PMS/PhOH system, the reaction was carried out in a 50-mL beaker containing 40 mL PhOH solution at varying concentrations; the pH of reaction solutions was maintained at 9.0 by adding 20 mM borate buffer. For $Co_3O_4$, $Co^{2+}$ was readily leached out in an acidic aqueous solution at low pH (Table S6). If $Co^{2+}$ dissolves into the reaction system, radical activation of PMS will occur, causing the utilization of PMS to decrease and degradation and mineralization of contaminants. This would lead to a decrease in the COD and TOC removal ratios. Therefore, for the $Co_3O_4$ system, DOTP is more efficient in alkaline conditions. Combined with the buffer range (basic pH) of borate, we chose pH 9.0 for the $Co_3O_4$ systems.

In a typical batch test, a dose of $Co_3O_4$ powder was added to the above solution. After 15 min ultrasonic dispersion, the suspension was stirred for 15 min to establish the adsorption-desorption equilibrium. Next, a dose of PMS was added to the suspension to initiate the reaction. To evaluate the decontamination performance, we studied the low-concentration system ([PhOH] = 12.5 mg $L^{-1}$ or 25 mg $L^{-1}$ (for TOC testing) and [catalyst] = 0.2 g $L^{-1}$) and high-concentration system ([PhOH] = 250 mg $L^{-1}$ (for COD testing) and [catalyst] = 1.0 g $L^{-1}$), and monitored the TOC and COD in the low-concentration and high-concentration systems, respectively.

To explore the universality of the process and the quantitative structure-activity relationship (QSAR), PhOH was replaced by several other organic pollutants, including BPA (low concentration: 30 mg $L^{-1}$, high concentration: 240 mg $L^{-1}$), AN (low concentration: 12.1 mg $L^{-1}$, high concentration: 242 mg $L^{-1}$), SA (low

concentration: 22.4 mg $L^{-1}$, high concentration: 280 mg $L^{-1}$), 4-CP (low concentration: 17 mg $L^{-1}$, high concentration: 340 mg $L^{-1}$), 2,6-M-PhOH (low concentration: 16 mg $L^{-1}$, high concentration: 288 mg $L^{-1}$), MOP (240 mg $L^{-1}$), MHB (300 mg $L^{-1}$), and AAP (300 mg $L^{-1}$). In addition, $Co_3O_4$ was replaced by other catalysts, such as cheaper metal oxides (FeMnO and $CuO_x$) and carbonaceous materials (CNTs and biochar); these experiments were conducted under near-neutral pH (without buffers). Another oxidant, potassium persulfate (PDS), was also tested. In this work, experiments were conducted in triplicate, and error bars are expressed as the arithmetic mean ± standard deviations.

**Analytical methods**. The qualitative and quantitative analyses including the reaction progress tracking, TOC measurements, COD measurements, $^{14}C$ labeling experiments, hypothetico-deductive experiments, scaling-up experiments, galvanic cell experiments, analogy experiments (PhOH/KI), in situ EPR experiments, radical scavenging experiments, nucleophilic inhibition experiments, KIE experiments, real environmental water anti-interference experiments, and all the characterizations techniques (TOC, COD, SEM-EDS, TEM, STEM, XPS, XRD, FTIR, TGA-FTIR, LC-MS, GPC, MALDI-TOF-MS, NMR, and EPR) are available in the Supporting Information.

**Quantum chemical calculations**

*Computational details*. All DFT calculations were performed by using the periodic plane-wave-based pseudo-potential method implemented in the Vienna ab initio simulation package (VASP)[53,54]. The core electron interaction was described by the projector-augmented wave (PAW) pseudopotential method[55,56]. The electron exchange and correlation energies contributed by valence electrons were treated by generalized gradient approximation in the Perdew–Burke–Ernzerhof functional with spin polarization[57]. The corresponding energy cutoff of the plane-wave basis was set up to 500 eV. $K$-point mesh was set to $3 \times 3 \times 1$ for slab ($4 \times 4 \times 4$ for crystal), following the Monkhorst–Pack procedure. Electron smearing was used via the Gaussian smearing technique with a smearing width of 0.05 eV. The geometry optimization was done until forces on each atom converged under 0.02 eV $Å^{-1}$. All kinetic energy barriers were calculated by using the climbing image nudged elastic band method (CI-NEB) with a maximum force of 0.05 eV $Å^{-1}$[58]. The van der Waals correction (D3) was also included throughout the DFT calculations[59].

*Models*. The calculated lattice constant of cubic $Co_3O_4$ crystal is 8.091 Å with an error of 0.3% when compared to its experimental value (8.065 Å). The most stable facet, $Co_3O_4$-001, was used to construct the slab for the representation of the catalyst surface. All the slabs consisted of seven atomic layers and were separated by a vacuum space of 20 Å. During structure optimizations, the bottom four layers of the slab were fixed at their bulk phase position, while the top three layers were fully relaxed. Different adsorption sites for PMS and phenol were tested to find the most probable adsorption configuration. The dipole correction was also added for the adsorption models. For the comparison system without a catalyst surface (PhOH + PMS), the same cell ($a = 11.443$ Å, $b = 11.443$ Å, $c = 26.271$ Å) was used to simulate the entire reaction process.

## Data availability

All data are available from the corresponding authors upon reasonable request.

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

## Acknowledgements

This work was supported by the National Natural Science Foundation of China (51821006, 21806160, and 21590812 by Prof. Yu). The numerical calculations in this work were conducted on the supercomputing system in the Supercomputing Center of the University of Science and Technology of China. The LC-MS measurements in this work were conducted in the National Demonstration Center for Experimental Chemistry Education (University of Science and Technology of China).

## Author contributions

Y.-J.Z. came up with the original idea; H.-Q.Y., J.-J.C., and M.E. supervised the project; Y.-J.Z., G.-X.H., J.-J.C., H.-Q.Y., and M.E. designed the experiments; Y.-J.Z. performed the experiments; Y.-J.Z., L.-L.T., S.-C.M., Z.Z., F.C., and Z.-Y.G. performed the characterizations; R.J. and Y.-Z.Y. helped with the data interpretations; Y.-J.Z., G.-X.H., X.-W.L., W.-W.L., L.R.W., H.-Q.Y., and M.E. wrote the manuscript; all authors commented on the manuscript.

## Competing interests

The authors declare no competing interests.
