## [Peer Review File · Nature Communications]

Title: Simultaneous Nanocatalytic Surface Activation of Pollutants and Oxidants for Highly Efficient Water DecontaminationREVIEWER COMMENTS

Reviewer #1 (Remarks to the Author):

This work proposed direct oxidative transfer process induced by the mixtures of cobalt oxides and persulfate and elucidated the mechanisms underlying the degradation of aromatic compounds. However, my main concern lies in the novelty of the findings presented in this study, though I admitted that the experiments were systemically performed and the results were very straightforward. The proposed process have been researched a lot in the previous works focused on non-radical persulfate activation (the name is different but the key mechanism is the same) and the approaches to characterize the process were quite similar, although this study made them more sophisticated. Accordingly, I do not believe that this work deserves publication in nature communications.

1. I am very doubtful about the novelty of this work. The main subject of this work is nothing but non-radical persulfate activation, which has been suggested to proceed via three different manners, i.e., mediated electron transfer, singlet oxygenation, and oxidation by high-valent metals (particularly when persulfate activators are based on transition metals such as cobalt and iron). In especial, the mediated electron transfer involving reactive PMS or PDS complexes is almost identical to the concept of “direct oxidative transfer process (DOTP)” proposed in this study. The characteristics of the process suggested here (for example, the evolution of polymeric products, selective reactivity proven based on the extent of the degradation of various organic compounds and QSAR analysis) were already well-established. This work simply repeated what have been done with diverse metal- and carbon-based materials previously except that the scope covered here was broader and the experimental approaches were slightly more refined.

2. I am not sure of the suitability of the terminology, i.e., activation in the title. The process of activation typically involves the improvement in reactivity. The oxidants that the authors mentioned in the title indicate PMS and PDS? How could pollutants be activated during PMS activation? As mentioned in the first comment, this study deal with non-radical PMS activation, although “direct oxidative transfer process” as the seemingly new terminology was used instead. The authors should show what they truly focused on in this study in a more concrete way.

3. Nearly all experiments in this work were performed primarily by PMS. As the authors are aware, Co₃O₄ is preferentially reactive toward PMS and has been demonstrated to generate sulfate radicals from PMS. On the other hand, cobalt-based activators barely interact with PDS. Relying on the hypothetic role of peroxides, PDS should have been more proper choice because the use of PDS simply avoids the possibility of radical generation.

4. The authors showed TOC and COD reduction after phenol oxidation by Co₃O₄/PMS. As the authors noted, in the heterogeneous catalytic oxidation, the results obtained the measurements should be more carefully analyzed because the intermediates from phenol oxidation could be simply removed via adsorption. The scenario is quite plausible considering that non-persulfate activation has been demonstrated to decompose organics in a very selective way. Have the authors monitored CO₂ evolution as the direct evidence of mineralization during phenol oxidation?

5. As the authors are well aware, some papers reported the progress of organic oxidation by high-valent metals. How could the authors differentiate DOTP from the reaction pathway induced by Co(IV) based

on the experimental results?

6. The authors implied the potential practicability of DOTP as compared to the conventional AOPs. Some technical shortcomings mostly related to cost issue, raised in the main text, could be admitted. However, as the authors suggested in this study, DOTP caused much more selective oxidation as compared to the existing AOPs utilizing OH radicals with the substrate-independent reactivity, which reveals the inability of DOTP to oxidatively remove a broad range of organic contaminants (I do not believe that DOTP is capable of oxidation of aliphatic organic contaminants containing no conjugated double bonds or aromatic compounds having multiple electron-withdrawing substituents). Further, I am concerned about the potential toxicity of the polymeric products – chlorophenols sometimes undergo polymerization to transform into dioxin congeners.

Reviewer #2 (Remarks to the Author):

Advanced oxidation processes (AOPs) are often considered to remove micropollutants in water, despite the large input of energy or chemicals and potential toxic byproducts from incomplete mineralization of organic micropollutants. In this study, a new mechanism, the direct oxidative transfer process (DOTP), was proposed and verified. This is excellent. In addition, the paper is well organized, the results are interesting, and the authors have provided fresh knowledge on the DOTP. However, there are some inappropriate definitions or discussions in some places are not deep enough. A revision is needed before the current version can be published.

Here're the detailed comments that may help improve the quality of the manuscript:

- 1) Line 63-64, it is not clear how the electron-equivalent non-conservation was defined. While TOC removal is in %, the oxidant has a unit of mole/L. Not the same units here.
- 2) Line 68, DOTP was not properly defined, which is a key to this work. Although it is mentioned “Such a non-destructive removal process has the distinct advantages of complete pollutant removal, avoidance of toxic byproduct formation, and low oxidizing agent consumption compared to the existing AOP”, how is so? It is not clear how DOTP can bring these distinct advantages.
- 3) Line 415-416, Ksp only gives a hint if it is easy to dissolve or precipitate, usually having nothing to do with pH values. Please elaborate on how the pH value of 8.68 was selected.
- 4) Line 430, it is not clear what structure or activity descriptor has been used when those micropollutants were selected. What is the purpose of establishing the QSAR model?
- 5) Line 451, in the quantum chemistry calculation, please indicate if any solvation models have been considered and involved in the calculations.
- 6) Line 111-112, the COD transferred from solution to catalyst surface – this is “adsorption” either through physical or chemical interactions. Why here was not defined as adsorption?
- 7) Line 117, I feel confused about the term “oxidative transfer” – if it is an oxidative process, the pollutant will “transform” rather than “transfer”. Could you please further clarify this term or DOTP?
- 8) Line 148 and 149, the XPS and FTIR spectra, along with the extraction experiment results, suggest there's chemical adsorption of phenol on Co₃O₄ catalyst. Also, the polymerization reaction is not an

oxidative reaction, right?

9) Line 190, it is suggested that the authors should mention the QSAR model was built upon phenolic compounds and maybe DOTPs are only valid for this group of pollutants.

10) Line 233-234, the authors messed up with the reaction rate between PMS and phenol or between radicals and phenol. The former should be second-order kinetics, while the latter is usually pseudo-first-order kinetics. The former also occurs in typical AOPs.

11) Line 254-255, there are EPR spectra in Figs 4g-4h, and please specify which radicals they are. In Co/PMS system, singlet oxygen (1O_2) may play a role in the pollutant degradation.

12) Line 335-336, regeneration of catalysts via organic solvent elution is a very expensive process. How many cycles can the catalysts be reused anyway? And how to treat the organic solvent containing pollutants?

13) Line 349-350, it is hard to believe the Co based materials are cheaper than AC. Please extend this cost estimation.

14) Line 373-375, maybe the Co based catalyst is a unique one for surface coupling and polymerization, which may be initiated by radicals that are often used in organic polymerization chemistry.

15) In the DOTP process, CR or PR reactions proceed with activation, stabilization, and accumulation. It seems the products on the catalyst surface are even more difficult to be mineralized than their parent phenolic compounds. Please comment on how DOTP process can be beneficial for TOC removal or mineralization.

Reviewer #3 (Remarks to the Author):

This manuscript reports on a brand-new mechanism on persulfate-based heterogeneous catalytic oxidation systems with nanosized metallic oxides, named the direct oxidative transfer process (DOTP). Differing from conventional advanced oxidation processes (AOPs), DOTPs can efficiently remove pollutants in the heterogeneous catalytic oxidation environment with the oxidizing agents serving as electron acceptors instead of generating highly reactive radical oxygen species. Meanwhile, DOTPs showed a high degradation and mineralization of micropollutants with much lower dosage of oxidizing agents than conventional AOPs. The manuscript was well-written, and the new mechanism was well verified by complete and logical experiments. The present study does show meaningful results on water purification. This paper can be published after carefully addressing the following issues.

1. The authors make an exciting statement on the mechanism of persulfate-based pollutants removal systems, yet the experimental results were significantly different from current reports in the similar Co_3O_4 /PMS/phenol (PhOH) system (generally accepted as radical dominated process). Even the authors claimed that the batch experiments were conducted under the condition of pH 9 with the addition of borate buffer. The determining factors, leading to DOTPs instead of radicals or other process, should be explained.

2. The DOTPs, occurred in the surface of catalysts with the direct redox reaction between pollutants and oxidizing agents, was close to the catalyst-mediated electron transfer mechanism. The differences need more discussions.

3. For Figure S35, further explanation was needed for that the reaction process was promoted with the addition of methanol and tert-butyl alcohol.

4. For ^{14}C -PhOH isotope labeling experiment, all the labeled carbon remained in the suspension after the reaction. The state of the ~20% removed COD need to be clarified.

5. The low cost of DOTPs was showing the potential on sustainable water purification. Have the calculation cover the cost of regeneration and waste disposal?

Response to Reviewers: NCOMMS-22-03683

Reviewer #1's comments

This work proposed direct oxidative transfer process induced by the mixtures of cobalt oxides and persulfate and elucidated the mechanisms underlying the degradation of aromatic compounds. However, my main concern lies in the novelty of the findings presented in this study, though I admitted that the experiments were systemically performed and the results were very straightforward. The proposed process have been researched a lot in the previous works focused on non-radical persulfate activation (the name is different but the key mechanism is the same) and the approaches to characterize the process were quite similar, although this study made them more sophisticated. Accordingly, I do not believe that this work deserves publication in nature communications.

1. I am very doubtful about the novelty of this work. The main subject of this work is nothing but non-radical persulfate activation, which has been suggested to proceed via three different manners, i.e., mediated electron transfer, singlet oxygenation, and oxidation by high-valent metals (particularly when persulfate activators are based on transition metals such as cobalt and iron). In especial, the mediated electron transfer involving reactive PMS or PDS complexes is almost identical to the concept of “direct oxidative transfer process (DOTP)” proposed in this study. The characteristics of the process suggested here (for example, the evolution of polymeric products, selective reactivity proven based on the extent of the degradation of various organic compounds and QSAR analysis were already well-established. This work simply repeated what have been done with diverse metal- and carbon-based materials previously except that the scope covered here was broader and the experimental approaches were slightly more refined.

Before replying to the specific comment of the reviewer, we would like to highlight the main revisions that we have made:

- The important clues/insights for better understanding the core of our work, such as the electron equivalent non-conservation contradiction in the field of heterogeneous catalytic persulfate oxidation, have been elaborated **and the novelty (i.e., DOTP compared to the existing AOP or adsorption processes) has been highlighted and clarified in the *Introduction* section.**
- A description about the carbon balance calculations has been provided and the fundamental differences between the DOTP and the mediated electron transfer pathway in AOPs have been clarified in detail.
- Additional evidences, including results of hypothetico-deductive experiments with the FeMnO-catalyzed PMS oxidation system, have been provided to verify that the surface coupling and polymerization pathways in DOTP can be generalized for different catalysts, rather than limited to the Co-based catalyst.

- The Co^{2+} leaching amount of Co_3O_4 under acidic, neutral, and basic pH conditions has been monitored by ICP-MS to address the rationale of conducting the experiments mainly at pH 9.0.
- The imprecise description of the pseudo-first-order and second-order kinetics in the *Results and Discussion* section and the misuse of $\text{Co}(\text{OH})_2$ K_{sp} in the *Methods* section in the previous manuscript have been corrected.

Our Specific Response: We appreciate the valuable time and efforts of the reviewer in reviewing our manuscript. We fully understand the reviewer's concern, but respectfully disagree with the reviewer's opinion about the novelty of this work. Because the new mechanism identified in this work may appear to be somewhat similar to that reported in previous works, we believe that we need to improve the clarity of the presentation of the paper to avoid the misunderstanding of the novelty of the paper by the reviewer. Herein, we would like to take this opportunity to fully explain the novelty and significance of our work in more detail.

Just as a note before our discussion, Reviewers 2 and 3 fully appreciated the novelty of the work. Specifically:

Reviewer 2: "In this study, **a new mechanism**, the direct oxidative transfer process (DOTP), was proposed and verified. **This is excellent**. In addition, the paper is well organized, **the results are interesting**, and the authors have provided **fresh knowledge on the DOTP**".

Reviewer 3: "This manuscript reports on a **brand-new mechanism** on persulfate-based heterogeneous catalytic oxidation systems with nanosized metallic oxides, named the direct oxidative transfer process (DOTP). The manuscript was well-written, and **the new mechanism was well verified by complete and logical experiments**. The present study does show **meaningful results on water purification**".

As shown in **Figure R1**, here we report a new catalytic process and mechanism differing fundamentally from all the existing ones. As mentioned by the reviewer, in the field of persulfate-based heterogeneous catalytic oxidation, it is generally accepted that the removal of pollutants is accomplished by an AOP, for which four different mechanisms of pollutant degradation/mineralization have been reported. These include free radicals, mediated electron transfer, singlet oxygenation, and high-valent metals mechanisms (the latter three are collectively referred to as non-radical mechanisms). However, after a thorough investigation and literature review, we have found that there is a **contradiction** among the reported data: electron equivalent balances cannot be explained with the above-mentioned mechanisms. As shown in **Supplementary Table 1**, the electron equivalent calculated from the actual dosage of oxidant (PMS or PDS) was much less than the electron equivalent calculated from the TOC removal efficiency (via mineralization), which is in contrast to the existing catalytic oxidation theory that the amount of oxidant should be in excess or at least stoichiometrically equivalent to that of pollutant. Such an unusual phenomenon actually exists widely in the persulfate-based heterogeneous catalytic literature and cannot be explained by the above-

mentioned four mechanisms. These results imply that a considerable fraction of pollutants is removed without consuming PMS through some unknown pathways. In addition, we notice that different catalytic mechanisms have been reported even in highly-similar PMS-based catalytic systems. For example, in the Fe-Mn oxide/PMS oxidation system, the controversy over the radical mechanism and the non-radical mechanism has always existed (Yao et al., *J. Hazard. Mater.*, 2014, 270, 61-70; Huang et al., *Environ. Sci. Technol.*, 2017, 51, 12611-12618; Du et al., *Chem. Eng. J.*, 2019, 376, 119193; Huang et al., *Environ. Sci. Technol.*, 2019, 53, 12610-12620; Yang et al., *Environ. Sci. Technol.*, 2020, 54, 3714-3724; Yang et al., *Chem. Eng. J.*, 2022, 429, 132280). Therefore, **controversy** regarding the PMS activation mechanisms and the activity origins remain.

To answer these two fundamental questions (i.e., electron equivalent imbalance contradiction and mechanism controversy), we provide a systematic and in-depth investigation into the persulfate-based heterogeneous catalytic decontamination processes. As shown in **Figure 1**, through comparatively investigating a variety of catalytic systems (different catalysts, such as Co_3O_4 , FeMnO , CuO , biochar, and CNT; various pollutants and oxidants, PMS and PDS), we discover a new pollutant removal pathway that differs fundamentally from all the reported degradation/mineralization pathway (i.e., the AOP). Here, we name it **direct oxidative transfer process (DOTP)**, where the pollutants were removed via a non-decomposing pathway that involves surface oxidation and accumulation on the catalyst. Specifically, in the DOTP nearly all the pollutants were removed from water via oxidative transfer (i.e., ~100% of TOC was enriched on the catalyst surface). Subsequently, we comprehensively elucidated the reaction pathway, structure-activity relationship, reaction mechanism, reaction thermodynamics and kinetics, and practical utility of such a DOTP, which all differ from those of the previously-recognized AOPs. Overall, instead of repeating a previously reported non-radical mechanism, our work reveals the working principles of widely studied persulfate-based heterogeneous catalytic oxidation systems to resolve outstanding contradictions and controversies in the field, and develops a new water decontamination technology.

Figure R1. Schematic diagram of the innovation of this work

Although some features of the DOTP (e.g., the direct oxidation (DO) in DOTP) may appear to be similar to the previously reported nonradical advanced oxidation mechanism (e.g., mediated electron transfer mechanism in AOP), **they are significantly different in essence**. The previously reported mechanisms still categorize the persulfate-based heterogeneous catalytic oxidation as an AOP, which cannot explain the above-mentioned remaining scientific questions in this field. Conversely, our discovery of the DOTP provides two important missing pieces of the puzzle, and fundamentally enriched the surface catalytic theory. **This is the most innovative aspect and major scientific contribution of our work.**

Moreover, the direct oxidation (DO) mechanism in this work is also substantially different from the previously reported mediated electron transfer mechanism. As shown in **Figure R2**, according to previous reports, the mediated electron transfer mechanism may involve two different scenarios, namely, electrons are transferred first from the pollutant to the catalyst, and then from the catalyst further to PMS/PDS (Type 1); electrons are transferred from the pollutant to the activated PMS/PDS-catalyst complex (Type 2) (Yun et al., Environ. Sci. Technol., 2018, 52, 7032-7042 and Ren et al., Environ. Sci. Technol., 2019, 53, 14595-14603). By contrast, in our DO mechanism the catalyst surface first activates the pollutant and the oxidant (PMS/PDS), respectively, and then a direct 2-electron redox reaction occurs between the activated pollutant and oxidant

molecules on the catalyst surface (the results of kinetics, galvanic cell experiments, DFT calculations, etc. **Lines 232-268, Figures 4a-4i and 5a-5f, Supplementary Figures 25-45**). In addition, the surface-stabilized pollutant intermediates and the pathways of surface coupling and polymerization (**Figure R1**) are also different from those of the mediated electron transfer processes.

Figure R2. Main differences between the DO mechanism in DOTP and the previously reported mediated electron transfer mechanism.

As for the evolution of polymeric products, the selectivity of pollutants, and the QSAR analysis mentioned by the reviewer, our results are also different from the previous reports. For example, the reported polymerization reactions so far are all ascribed to radical polymerization processes (such as the polymerization of phenoxy radicals) (Guan et al., *Environ. Sci. Technol.*, 2017, 51, 10718-10728; Liu et al., *Appl. Catal., B* 2019, 254,166-173; Zhang et al., *Chem. Eng. J.*, 2020, 397, 125351). In contrast, in the DOTP reaction of our work, the catalyst surface catalyzes the direct redox reaction between the pollutant and oxidant to generate the surface-stabilized pollutant intermediates for further surface coupling or polymerization, in which process no radicals are involved. Moreover, the DOTP includes both surface polymerization and surface coupling (to generate diphenoquinone products, rather than oligomers) pathways.

In terms of the selectivity for pollutant removal, in this work we revealed the origin of selectivity based on the reaction mechanism, and the specific reaction sites of pollutant structure. In contrast, previous reports simply provide a preliminary summary of the experimental results regarding the removal efficiency of different pollutants (e.g., phenol vs benzoic acid) (Hu et al., *Environ. Sci. Technol.*, 2017, 51, 11288-11296 and Zhu et al., *Environ. Sci. Technol.*, 2019, 53, 307-315).

As for the QSAR analysis, most previous reports mainly focus on the relationship between the oxidation rate and the chemical properties (such as chemical potential) of

different pollutants (Arnold et al., *Environ. Sci.: Processes Impacts* 2017, 19, 324–338 and Ren et al., *Environ. Sci. Technol.*, 2019, 53, 14595-14603). In contrast, in our work, we establish more detailed relationships between the molecular structure of pollutants and their reaction pathways, product solubility, and oxidant consumption based on the analysis of the DOTP process and mechanism. With this QSAR analysis, we can make judgments and predictions from multiple aspects (i.e., the reaction pathways, the product solubility, and the oxidant consumption) simply based on the molecular structure of pollutants, providing a valuable reference to guide practical applications. Meanwhile, we can also verify the reliability of the overall mechanism analysis of the DOTP via the QSAR analysis. Therefore, the QSAR analysis also contributes substantial novelty to the field of persulfate-based heterogeneous catalytic pollutant removal.

Overall, our work with diverse metal oxides and carbon materials is not a repetition of the previous studies. Considering that these representative materials (i.e., Co_3O_4 , FeMnO , CuO , biochar, and CNT) have received great attention in the field of persulfate-based heterogeneous catalytic oxidation (Zeng et al., *Environ. Sci. Technol.*, 2015, 49, 2350-2357; Huang et al., *Environ. Sci. Technol.*, 2017, 51, 12611-12618; Zhang et al., *Environ. Sci. Technol.*, 2014, 48, 5868-5875; Huang et al., *J. Mater. Chem. A*, 2018, 6, 8978; Yun et al., *Environ. Sci. Technol.*, 2018, 52, 7032-7042), we deliberately chose them to verify the universality of the DOTP. Our results provide convincing evidences to resolve the previous common misconceptions in this field.

2. I am not sure of the suitability of the terminology, i.e., activation in the title. The process of activation typically involves the improvement in reactivity. The oxidants that the authors mentioned in the title indicate PMS and PDS? How could pollutants be activated during PMS activation? As mentioned in the first comment, this study deal with non-radical PMS activation, although “direct oxidative transfer process” as the seemingly new terminology was used instead. The authors should show what they truly focused on in this study in a more concrete way.

The oxidants in the title refer to PMS and PDS. In the DOTP, the pollutant and oxidant are activated respectively by the catalyst surface before the DO reaction occurs. The activation of both pollutant and oxidant are reflected from two aspects:

1) Changes in the chemical potential (as shown in the mechanism description section of **Figure 6**). Before activation, the pollutant and oxidant in the solution cannot react with each other due to their small chemical potential difference. Once activated on the catalyst surface, the chemical potential difference between the pollutant and oxidant is increased, enabling the reaction to occur.

2) Changes in the orbital electron cloud (**Figure R2**). Before activation, the electron clouds of the pollutant and oxidant in the solution are relatively concentrated and not delocalized, making it difficult for electrons to flow from pollutant to oxidant (i.e., redox reaction). After activation on the catalyst surface, both the electron clouds of the pollutant and oxidant are delocalized to a certain degree. As shown in

Supplementary Figure 40a and 40b, both the orbital electron clouds of the pollutant and the oxidant become partially fused with that of the catalyst surface, which favors the contact between the delocalized electron clouds of the pollutant and oxidant to enable electron transfer to occur, i.e., activation to promote a direct redox reaction.

3. Nearly all experiments in this work were performed primarily by PMS. As the authors are aware, Co_3O_4 is preferentially reactive toward PMS and has been demonstrated to generate sulfate radicals from PMS. On the other hand, cobalt-based activators barely interact with PDS. Relying on the hypothetical role of peroxides, PDS should have been more proper choice because the use of PDS simply avoids the possibility of radical generation.

Many thanks for the reviewer's suggestion to replace the $\text{Co}_3\text{O}_4/\text{PMS}$ system with the $\text{Co}_3\text{O}_4/\text{PDS}$ system. In fact, there are several reasons why we chose the $\text{Co}_3\text{O}_4/\text{PMS}$ system rather than the $\text{Co}_3\text{O}_4/\text{PDS}$ system in this work. As mentioned above, our work demonstrates that, in the premise of strictly heterogeneous catalysis, the main reaction pathway in the $\text{Co}_3\text{O}_4/\text{PMS}$ system is the DOTP, rather than the previously reported sulfate radical-based AOP. In addition to the $\text{Co}_3\text{O}_4/\text{PMS}$ system, other heterogeneous persulfate oxidation systems are also dominated by DOTPs rather than AOPs. This sheds light on the previous misunderstandings of AOPs in the entire field. Since the $\text{Co}_3\text{O}_4/\text{PMS}$ system was previously considered to be a typical free-radical based AOP system and has been studied more extensively than the $\text{Co}_3\text{O}_4/\text{PDS}$ system, we deliberately chose it as a case study to clarify the misunderstandings. Then, combined with the expansion and verification of other heterogeneous catalytic systems, the results would be more convincing.

4. The authors showed TOC and COD reduction after phenol oxidation by $\text{Co}_3\text{O}_4/\text{PMS}$. As the authors noted, in the heterogeneous catalytic oxidation, the results obtained the measurements should be more carefully analyzed because the intermediates from phenol oxidation could be simply removed via adsorption. The scenario is quite plausible considering that non-persulfate activation has been demonstrated to decompose organics in a very selective way. Have the authors monitored CO_2 evolution as the direct evidence of mineralization during phenol oxidation?

The main phenol removal pathway in the $\text{Co}_3\text{O}_4/\text{PMS}$ system is DOTP. With the TOC/COD removal test combined with the quantitative and qualitative analysis (^{14}C labeling, EDX and mapping, and TGA) of organic matter on the catalyst surface (**Figures 1a-1f and 2a-2b, etc.**), we have validated that phenol in the $\text{Co}_3\text{O}_4/\text{PMS}$ system was completely (~100%) removed from water via non-decomposing oxidation and accumulation on the catalyst surface (proven by the carbon balance calculation), rather than being degraded and mineralized to CO_2 .

From the elution of organics from the reacted catalysts, we have proven that the substances accumulated on the catalyst surface are derived from phenol coupling/polymerization, not from phenol oxidation and intermediates adsorption, because in the latter case the surface deposit cannot be observed by TEM and typically

can be easily eluted by organic solvents. Our results show that, after the reaction in the phenol/Co₃O₄/PMS system, the organic substances on the catalyst surface could not be eluted by solvents including ethanol, chloroform, toluene, and acidic and alkaline aqueous solutions. In addition, obvious coating layers on the catalyst surface were observed by TEM. The pyrolysis temperature range of these substances was 300 ± 20 °C. These results are all consistent with the properties of organic polymers rather than the intermediates from phenol oxidation.

5. As the authors are well aware, some papers reported the progress of organic oxidation by high-valent metals. How could the authors differentiate DOTP from the reaction pathway induced by Co(IV) based on the experimental results?

To the best of our knowledge, there is no report to show the existence of Co(IV) in the Co₃O₄/PMS system. In addition, pollutants removal has been found to occur exclusively via oxidative degradation pathway in all the reaction systems involving Co(IV) mechanism, e.g., homogeneous Co²⁺/PMS (Zong et al., *Environ. Sci. Technol.* 2020, 54(24), 16231-16239; Liu et al., *J. Hazard. Mater.* 2021, 416, 125679; Zong et al., *Appl. Catal., B* 2022, 300, 120722); homogeneous Co(II)/peracetic acid (Liu et al., *Water Res.* 2021, 201, 117313); heterogeneous cobalt phthalocyanine/H₂O₂ (Li et al., *Appl. Catal., B* 2015, 163, 105-112; Li et al., *Appl. Surf. Sci.* 2018, 434, 1112-1121); and heterogeneous Co(II)-doped g-C₃N₄/peracetic acid (Liu et al., *Environ. Sci. Technol.* 2021, 55(18), 12640-12651). In contrast, our work reveals a non-decomposing oxidative transfer pathway (i.e., DOTP, including the surface coupling and polymerization reaction) of the pollutants in diverse heterogeneous persulfate oxidation systems (**Figures 1-3 and Table 1**), which is fundamentally different from the Co(IV)-induced reaction pathway.

6. The authors implied the potential practicability of DOTP as compared to the conventional AOPs. Some technical shortcomings mostly related to cost issue, raised in the main text, could be admitted. However, as the authors suggested in this study, DOTP caused much more selective oxidation as compared to the existing AOPs utilizing OH radicals with the substrate-independent reactivity, which reveals the inability of DOTP to oxidatively remove a broad range of organic contaminants (I do not believe that DOTP is capable of oxidation of aliphatic organic contaminants containing no conjugated double bonds or aromatic compounds having multiple electron-withdrawing substituents). Further, I am concerned about the potential toxicity of the polymeric products – chlorophenols sometimes undergo polymerization to transform into dioxin congeners.

Regarding the application scope, we think that the DOTP is mainly suitable for aromatic organic pollutants with electron-rich structures (e.g., phenols, amines and thiophenes). Although radical-based AOPs can deal with a broader range of organic contaminants, selective decontamination processes are more advantageous in many cases and may complement radical-based AOPs to realize higher treatment efficiency and economic benefits. As a selective oxidation technology for targeted pollutants,

DOTP still has a broad application scope because phenols, amines, and other types of aromatic organic pollutants with electron-rich structures are widely present in wastewaters from various industries such as pharmaceuticals manufacturing, petrochemicals, chemical synthesis, papermaking, dye manufacturing, pesticide plants, coking plants, organic chemical raw materials manufacturing, and non-ferrous metal smelting, etc. For example, the treatment of emerging micropollutants, such as bisphenol A (BPA), 4-chlorophenol (4-CP), and sulfamethoxazole (SMZ) (Alsaiee et al., *Nature*, 2016, 529, 190-194, Peter et al., *Acc. Chem. Res.* 2019, 52, 605-614, and Mailler et al., *Sci. Total. Environ.* 2016, 542, 983-996), has recently been a primary focus in the field of water purification.

In DOTP, the chlorophenols products are enriched onto the catalyst surface by non-decomposing oxidative transfer, so that they are completely removed from the water. Thus, the process will leave no toxic products in water. For the catalyst with accumulated DOTP products after the reaction, benign treatment and effective regeneration of the catalyst through calcination combined with exhaust gas treatment could be adopted (**Figure 2b and 2c, and Supplementary Figure 47**).

Reviewer #2's comments

Advanced oxidation processes (AOPs) are often considered to remove micropollutants in water, despite the large input of energy or chemicals and potential toxic byproducts from incomplete mineralization of organic micropollutants. In this study, a new mechanism, the direct oxidative transfer process (DOTP), was proposed and verified. This is excellent. In addition, the paper is well organized, the results are interesting, and the authors have provided fresh knowledge on the DOTP. However, there are some inappropriate definitions or discussions in some places are not deep enough. A revision is needed before the current version can be published.

The reviewer's valuable time and efforts on our manuscript are greatly appreciated. We have made revisions according to the reviewer's suggestions.

Before replying to the specific comment of the reviewer, we would like to highlight the main revisions that we have made:

- The important clues/insights for better understanding the core of our work, such as the electron equivalent non-conservation contradiction in the field of heterogeneous catalytic persulfate oxidation, have been elaborated **and the novelty (i.e., DOTP compared to the existing AOP or adsorption processes) has been highlighted and clarified in the *Introduction* section.**
- A description about the carbon balance calculations has been provided and the fundamental differences between the DOTP and the mediated electron transfer pathway in AOPs have been clarified in details.
- Additional evidences, including results of hypothetico-deductive experiments with the FeMnO-catalyzed PMS oxidation system, have been provided to verify that the surface coupling and polymerization pathways in DOTP can be generalized for different catalysts, rather than limited to the Co-based catalyst.
- The Co^{2+} leaching amount of Co_3O_4 under acidic, neutral, and basic pH conditions has been monitored by ICP-MS to address the rationale of conducting the experiments mainly at pH 9.0.
- The imprecise description of the pseudo-first-order and second-order kinetics in the *Results and Discussion* section and the misuse of $\text{Co}(\text{OH})_2$ K_{sp} in the *Methods* section in the previous manuscript have been corrected.

Here're the detailed comments that may help improve the quality of the manuscript:

1) Line 63-64, it is not clear how the electron-equivalent non-conservation was defined. While TOC removal is in %, the oxidant has a unit of mole/L. Not the same units here.

We are sorry that the expression in our submitted manuscript was not clear. The electron-equivalent non-conservation here means that, in the reaction process, the electron equivalent consumed by the oxidant, calculated according to the actual dosage

of PMS or PDS, was much less than that actually donated from the pollutant, calculated according to the mineralized organics (estimated based on the TOC removal ratio).

To address the reviewer's concern, we have revised the description of the electron-equivalent non-conservation in the revised manuscript (**Lines 61-66**: Another widely-overlooked inconsistency is the electron equivalent non-conservation contradiction in the reported PMS-based heterogeneous catalytic systems. Specifically, the electron equivalent consumed by the oxidant (calculated according to the actual dosage of PMS or PDS) is much lower than the donated electron equivalent from the pollutant (calculated according to the mineralized organics) (Table S1). This contradiction cannot be explained by the existing oxidation theory for pollutant degradation.) and revised the **Supplementary Table 1** to keep the units consistent.

2) Line 68, DOTP was not properly defined, which is a key to this work. Although it is mentioned "Such a non-destructive removal process has the distinct advantages of complete pollutant removal, avoidance of toxic byproduct formation, and low oxidizing agent consumption compared to the existing AOP", how is so? It is not clear how DOTP can bring these distinct advantages.

In the *Results and Discussion* section, we described the DOTP process in detail: the pollutants and the oxidants are both activated on the catalyst surface and then the pollutants are directly oxidized by the oxidants to form the pollutant intermediates, which are stabilized by the catalyst surface and undergo the surface coupling and polymerization reactions, and finally, the formed products are enriched on the catalyst surface. According to our experimental results (**Figures 1a-1f and 2b, and Lines 794-797**), the DOTP can transfer ~100% of the TOC from the water phase to the catalyst surface with a low dose of persulfate dosage (less than 2 times the molar concentration of the pollutants), so that the complete water purification can be achieved by separating the catalyst and water. Due to the circumvention of pollutant degradation process, no toxic degradation by-products will be formed. These descriptions are given in the *Results and Discussion* section. For the *Introduction* section, however, considering the space limitation and the absence of experimental data, these are briefly described without detailed explanation.

3) Line 415-416, K_{sp} only gives a hint if it is easy to dissolve or precipitate, usually having nothing to do with pH values. Please elaborate on how the pH value of 8.68 was selected.

We thank the reviewer for pointing out our misuse of K_{sp} . To determine the appropriate pH for the $\text{Co}_3\text{O}_4/\text{PMS}/\text{PhOH}$ system, we measured the Co^{2+} leaching and calculated the DOTP ratio of this system at different pHs. As shown in **Table R1**, at higher pH, less Co^{2+} was released. However, when the pH was increased from 9.1 to 10.0, the DOTP ratio declined from 97% to 87% (**Figure 1a and Supplementary Figure 7c**). Conjunctively considering the buffer range of borate buffer, we chose pH 9.0 in most experiments for the $\text{Co}_3\text{O}_4/\text{PMS}/\text{PhOH}$ system.

Table R1 | Co^{2+} leaching amount in the reaction of the $\text{Co}_3\text{O}_4/\text{PhOH}/\text{PMS}$ system

determined by ICP-MS.

pH	3.8	7.8	9.1	10.0
Co ²⁺ leaching (mg L ⁻¹)	0.359	0.005	0.003	0

To address the reviewer's concern, we have rewritten the description in the revised manuscript (**Lines 422-423**: For Co₃O₄, Co²⁺ was readily leached out in acidic aqueous solution at low pH (Table S6) **and 427**: Combined with the buffer range (basic pH) of borate, we chose pH 9.0 for the Co₃O₄ systems) and added the above data into the supporting information (**Supplementary Table 6**).

4) Line 430, it is not clear what structure or activity descriptor has been used when those micropollutants were selected. What is the purpose of establishing the QSAR model?

The structure descriptor in QSAR was the molecular structure of the pollutants, specifically, the number of active hydrogen sites (i.e., hydrogen atoms at the ortho and para positions of the electron-donating group on the benzene ring). The effect (activity) descriptors include the proportion of reaction pathways, product solubility from catalyst surface, and the oxidant consumption.

We established the QSAR for two purposes. Firstly, it can be used to verify the reliability of our analysis on the DOTP reaction process and mechanisms. In addition, it can provide valuable information to guide future pollution control applications, because with this information the optimal reaction conditions and process may be determined simply based on the molecular structure of pollutants.

5) Line 451, in the quantum chemistry calculation, please indicate if any solvation models have been considered and involved in the calculations.

The solvation models were not considered in this work. We mainly calculated the thermodynamic free energies and kinetic energy barriers in the elementary reaction processes on catalyst surface. In this case, the adsorption and desorption of the reaction molecules between the solution and the catalyst surface were not involved (Gong et al., J. Catal., 2019, 373, 322-335, Kattel et al., Science, 2017, 355, 1296-1299, and Zhao et al., Nat. Commun., 2020, 11, 2455).

6) Line 111-112, the COD transferred from solution to catalyst surface – this is “adsorption” either through physical or chemical interactions. Why here was not defined as adsorption?

Our analysis of the reaction pathway and mechanisms strongly support that the COD transfer process occurred here belong to DOTP, rather than the commonly-recognized adsorption process. These two processes differ fundamentally in several aspects: 1) The DOTP process originates from a surface-catalyzed oxidation reaction,

after which pollutant molecules are coupled/polymerized on the catalyst surface and directly enriched *in situ*, while adsorption process typically features direct transfer of pollutants or reaction products onto the material surface; 2) The Co_3O_4 barely adsorb PhOH in $\text{Co}_3\text{O}_4/\text{PMS}/\text{PhOH}$ system (**Figure 1c**), indicating a very limited adsorption capacity of Co_3O_4 which cannot explain the high pollutant removal performance in our study; and 3) The DOTP products enriched on the catalyst surface forms a “film” with a thickness of tens of nanometers (**Figure 2a**), which cannot occur for the traditional physical or chemical adsorption. For these reasons, while the process may be similar to an activated or reactive adsorption process, we think that since pollutant oxidation is inextricably involved in the pollutant transfer to the surface that it might be more appropriate to use DOTP rather than adsorption to describe our findings.

7) Line 117, I feel confused about the term “oxidative transfer” – if it is an oxidative process, the pollutant will “transform” rather than “transfer”. Could you please further clarify this term or DOTP?

The oxidative transfer process of DOTP can be divided into two steps. In the first step, pollutants are directly oxidized by oxidant on the catalyst surface; in the second step, the stabilized pollutant intermediates (i.e., the surface oxidation products of pollutants) on the surface undergo coupling and polymerization reactions and then the products directly accumulate on the catalyst surface. The “oxidative” and “transfer” in DOTP reflect the oxidation process of the pollutants and the accumulation result of the pollutants from water onto the catalyst surface, respectively. “Transform” can summarize the first step only, but fails to include the second step. Therefore, we think the word “transfer” can describe the process here more accurately than “transform”.

8) Line 148 and 149, the XPS and FTIR spectra, along with the extraction experiment results, suggest there’s chemical adsorption of phenol on Co_3O_4 catalyst. Also, the polymerization reaction is not an oxidative reaction, right?

The results of the adsorption control system in **Figure 1c** and the adsorption process before PMS addition in **Figure R3**, show that Co_3O_4 can barely adsorb PhOH. As for the products enriched on the catalyst surface after the reaction, we proved that they are not the adsorbed small molecules (including phenol), but mainly the cross-linked polymers. This observation is supported by the light contrast layers in the TEM images, the TGA decomposition temperature range, and the fact that it cannot be eluted with various solvents (HCl, NaOH, ethanol, chlorophenol, and toluene). In addition, the valence bond information (C-O-C, C-C and C-O) in the XPS and FTIR spectra of the products further indicates that the cross-linked polymer should be a polyphenylene ether structure. Furthermore, we confirm the products structure via the hypothetico-deductive experiments by using 2,6-dimethylphenol (2,6-M-PhOH) as the pollutant molecule.

With the analysis of the DOTP reaction process, we illustrate that the polymerization reaction occurs immediately after the direct redox reaction between

pollutants and oxidants on the catalyst surface. The oxidation reaction is critical to drive the stabilization of pollutant intermediates on the catalyst surface and their subsequent coupling and polymerization reactions. Therefore, the surface polymerization reaction from PhOH to polyphenylene ether is a type of oxidative reaction. Besides, judging from the molecular structures of PhOH and the polymerization product (e.g., $2 \text{ PhOH} \rightarrow \text{polyphenylene ether}$), the average valence state of C in the polymerization product is higher than that in PhOH (PhOH , $-2/3$ and $\text{polyphenylene ether}$, $-1/2$). This also supports that the polymerization reaction is an oxidative one.

Figure R3 | Adsorption and DOTP reaction processes for PhOH removal in the $\text{Co}_3\text{O}_4/\text{PMS}/\text{PhOH}$ system ($[\text{PhOH}] = 12.5 \text{ mg L}^{-1}$, $[\text{Co}_3\text{O}_4] = 0.2 \text{ g L}^{-1}$, $[\text{PMS}]: [\text{PhOH}] = 2$, $\text{pH} = 9$, borate buffer = 20 mM).

9) Line 190, it is suggested that the authors should mention the QSAR model was built upon phenolic compounds and maybe DOTPs are only valid for this group of pollutants.

Accepting the reviewer's suggestion, we have revised the description of the pollutants type in QSAR model (**Lines 192-193: Quantitative structure–activity relationship analysis of DOTP reactions with phenols**). As shown in **Figure 1h and Supplementary Figure 11**, we also reveal the universality of DOTP towards amines pollutants (i.e., anilines and sulfonamides). Combined with the analysis of the DOTP mechanism and thermodynamics, it is expected that other pollutants with electron-donating groups (e.g., $-\text{SH}$) should also undergo a similar DOTP. Therefore, the DOTP may apply to diverse pollutants beyond phenolics.

10) Line 233-234, the authors messed up with the reaction rate between PMS and phenol or between radicals and phenol. The former should be second-order kinetics, while the latter is usually pseudo-first-order kinetics. The former also occurs in typical AOPs.

We agree with the reviewer that the second-order kinetics also exists in some AOPs, especially in some cases where oxidant molecules directly oxidize pollutants or where

DOTPs in the heterogeneous catalytic systems are mistaken for AOPs (Yang et al., Environ. Sci. Technol. 2018, 52, 5911-5919 and Chen et al., Environ. Sci. Technol. 2018, 52, 1461-1470). If it is an AOP, in which pollutants are degraded by the generated reactive oxygen species (ROS), pseudo-first-order kinetics usually applies because the reaction rate of ROS and pollutants in such process is often much larger than the ROS production rate and hence the steady-state ROS concentration can be treated as a constant. However, in the DOTP process, phenol was directly oxidized by PMS on the catalyst surface, so it tends to be a second-order kinetics.

To address the reviewer's concern, we have modified the descriptions in the revised manuscript (**Lines 235-237**: At a PMS to PhOH dosage ratio of 1.8:1, the reaction followed second-order kinetics (Fig. **4c**) rather than pseudo-first-order kinetics (Fig. **S25**); the latter is typical for conventional ROS-based AOPs).

11) Line 254-255, there are EPR spectra in Figs 4g-4h, and please specify which radicals they are. In Co/PMS system, singlet oxygen ($^1\text{O}_2$) may play a role in the pollutant degradation.

Thanks for the reviewer's suggestion. We specified the type of radicals in the caption of **Figure 4 (Lines 848-854)**: Differing from the typical radical systems (i.e., $\bullet\text{OH}$ in the $\text{Fe}^{2+}/\text{H}_2\text{O}_2/\text{PhOH}$ system and $\text{SO}_4^{\bullet-}$ in the ZVI/PMS/PhOH system) that exhibited DMPO- OH^{\bullet} (with hyperfine couplings $a_{\text{N}} = a_{\beta\text{-H}} = 14.9$ G) or DMPO- $\text{SO}_4^{\bullet-}$ (with hyperfine splitting constants of $a_{\text{N}} = 13.2$ G, $a_{\beta\text{-H}} = 9.6$ G, $a_{\gamma\text{-H1}} = 1.48$ G and $a_{\gamma\text{-H2}} = 0.78$ G) signals in the EPR spectra (**h**), in the $\text{Co}_3\text{O}_4/\text{PMS}/\text{PhOH}$ system (**g**), only the DMPOX signal (with a 1:1:1 triplet ($a_{\text{N}} = 7.2$ G) of 1:2:1 triplet ($a_{\gamma\text{-H}} = 4.1$ G, 2H)) appeared from the beginning of the reaction and then went through a process of initial increasing and later decreasing. In addition, no singlet oxygen signal peaks appeared in **Figure 4g**.

To address the reviewer's concern, we have added the labeling of the radical types in the revised manuscript (**Figures 4g-4h**).

12) Line 335-336, regeneration of catalysts via organic solvent elution is a very expensive process. How many cycles can the catalysts be reused anyway? And how to treat the organic solvent containing pollutants?

The lifetime of the catalyst is affected by the pollutant concentration and the catalyst can be used until the catalyst reaches saturation. As shown in **Supplementary Figures 46 and 47**, the catalysts can be reused for at least 15 cycles when the initial pollutant concentration is at ~ 10 mg/L level; while in the lower-concentration environment, as shown by **Figure 7** and **Supplementary Figure 52**, the catalysts can be used for a much longer time period.

If the DOTP process is used to deal with a known reactant or produces a high-value reaction product, after the catalyst is eluted with an organic solvent, the product-containing eluate can be distilled to recover the product. For practical water treatment applications, when considering the cost, cheap catalyst should be a good choice. For

example, biomass may be used and burnt as fuel after the catalyst has become saturated.

13) Line 349-350, it is hard to believe the Co based materials are cheaper than AC. Please extend this cost estimation.

For the analysis of the DOTP process and mechanism, we used Co_3O_4 as the model catalyst, but in the section of *Evaluation of DOTP in practical applications*, we used the activated carbon (AC) as the DOTP catalyst because AC is convenient for the construction of fixed bed reaction column and for the scale evaluation. Therefore, the cost evaluation here is based on AC rather than Co_3O_4 .

To address the reviewer's concern, we have modified the descriptions in the revised manuscript (**Lines 349-352**: When compared with adsorption (physical process) using activated carbon as the representative adsorbent (Figs. **S48 and S49**, Fig. **7d and 7e**, and Table **S3**), the DOTP using activated carbon as the catalyst in a PDS oxidation system (AC/PDS) showed a relatively higher decontamination capacity and longer stability).

14) Line 373-375, maybe the Co based catalyst is a unique one for surface coupling and polymerization, which may be initiated by radicals that are often used in organic polymerization chemistry.

With detailed characterizations, including EPR capture of free radicals, radical-quenching experiments, and quantitative detection of free radicals, we have revealed that there is no participation and generation of free radicals in the tested Co_3O_4 /PMS reaction system. The surface coupling and polymerization pathways in DOTP are not initiated by free radicals, but by a $2e^-$ direct oxidation on the catalyst surface.

In addition, the surface coupling and polymerization pathways in DOTP are not confined to the Co_3O_4 /PMS system. In the MnFeO /PMS/2,6-MPhOH system, the reaction products enriched on the catalyst surface were dissolved in ethanol and toluene successively, and then the eluted products were identified by LC-MS, GPC and MALDI-TOF-MS, respectively. As shown in **Figure R4**, the same pathways of surface coupling and polymerization reaction were also identified in the MnFeO -catalyzed system. Therefore, the surface coupling and polymerization pathways in DOTP are not limited to the Co-based catalyst.

Figure R4. Reaction pathway analyses of DOTP in the FeMnO/PMS/2,6-MPhOH system. **a, b**, Liquid chromatogram (**a**) and mass spectrum (**b**) of the products washed off by ethanol. **c**, Schematic of coupling reaction pathway. **d, e**, Gel permeation chromatogram (**d**) and MALDI-TOF-MS spectrum (**e**) of the products washed off by toluene. **f**, Schematic of polymerization reaction pathway. PDI, polymer dispersity index; M_n , number-average molecular weight; M , theoretical value of the molar mass of the polymeric unit; N , degree of polymerization.

15) In the DOTP process, CR or PR reactions proceed with activation, stabilization, and accumulation. It seems the products on the catalyst surface are even more difficult to be mineralized than their parent phenolic compounds. Please comment on how DOTP process can be beneficial for TOC removal or mineralization.

In the DOTP process TOC is transferred from water to the catalyst surface. Then, complete water purification can be achieved via simply separating the catalyst from

water. The DOTP does not involve pollutant mineralization. Regarding its practical application (**Figures 7d-7e, Supplementary Table 3, and Supplementary Figure 49**), for example when granular activated carbon was used as DOTP catalyst in fixed-bed reactor, the water containing pollutants and oxidants may continuously flow through the reactor where the pollutants can be completely removed via DOTP. Then, the used granular activated carbon, with concentrated pollutants on the surface, may be burned as fuel. The entire process could be more cost-effective and efficient than direct mineralization of TOC in aqueous solution.

Reviewer #3's comments

This manuscript reports on a brand-new mechanism on persulfate-based heterogeneous catalytic oxidation systems with nanosized metallic oxides, named the direct oxidative transfer process (DOTP). Differing from conventional advanced oxidation processes (AOPs), DOTPs can efficiently remove pollutants in the heterogeneous catalytic oxidation environment with the oxidizing agents serving as electron acceptors instead of generating highly reactive radical oxygen species. Meanwhile, DOTPs showed a high degradation and mineralization of micropollutants with much lower dosage of oxidizing agents than conventional AOPs. The manuscript was well-written, and the new mechanism was well verified by complete and logical experiments. The present study does show meaningful results on water purification. This paper can be published after carefully addressing the following issues.

We thank the reviewer for the helpful comments to our work, and have made revisions according to the reviewer's suggestions.

Before replying to the specific comment of the reviewer, we would like to highlight the main revisions that we have made:

- The important clues/insights for better understanding the core of our work, such as the electron equivalent non-conservation contradiction in the field of heterogeneous catalytic persulfate oxidation, have been elaborated **and the novelty (i.e., DOTP compared to the existing AOP or adsorption processes) has been highlighted and clarified in the *Introduction* section.**
- A description about the carbon balance calculations has been provided and the fundamental differences between the DOTP and the mediated electron transfer pathway in AOPs have been clarified in details.
- Additional evidences, including results of hypothetico-deductive experiments with the FeMnO-catalyzed PMS oxidation system, have been provided to verify that the surface coupling and polymerization pathways in DOTP can be generalized for different catalysts, rather than limited to the Co-based catalyst.
- The Co^{2+} leaching amount of Co_3O_4 under acidic, neutral, and basic pH conditions has been monitored by ICP-MS to address the rationale of conducting the experiments mainly at pH 9.0.
- The imprecise description of the pseudo-first-order and second-order kinetics in the *Results and Discussion* section and the misuse of $\text{Co}(\text{OH})_2$ K_{sp} in the *Methods* section in the previous manuscript have been corrected.

1. The authors make an exciting statement on the mechanism of persulfate-based pollutants removal systems, yet the experimental results were significantly different from current reports in the similar Co_3O_4 /PMS/phenol (PhOH) system (generally accepted as radical dominated process). Even the authors claimed that the batch experiments were conducted under the condition of pH 9 with the addition of borate

buffer. The determining factors, leading to DOTPs instead of radicals or other process, should be explained.

We conducted the experiments of DOTP under diverse reaction conditions (including different solution pHs, concentrations of pollutants and dosages of catalyst and oxidant, **Supplementary Figures 7-10**) and in various catalytic systems (including various catalysts, pollutants and oxidants, **Figures 1g and 1h** and **Supplementary Figures 5, 6, 11 and 12**). The results show that the DOTP dominated the pollutant removal processes in all the tested systems, with a ratio of above 80%. Therefore, the DOTP is ubiquitous in the heterogeneous catalytic persulfate systems, and there are no special requirements for its occurrence except for the premise of strictly heterogeneous catalysis.

To reveal the previous misunderstandings (i.e., AOPs), we chose the $\text{Co}_3\text{O}_4/\text{PMS}$ system, for which a free radical mechanism is commonly recognized. It is important to point out that, if the Co_3O_4 has obvious Co^{2+} leaching in the solution, the homogeneous catalysis may also lead to free radical generation. We have demonstrated that the DOTP still accounted for the main pathway (ratio = 60%, **Supplementary Figure 7a**) in the $\text{Co}_3\text{O}_4/\text{PMS}$ system under the acidic conditions (pH=3.8). However, to avoid the interference of homogeneous catalysis on our mechanistic studies, we chose to operate the $\text{Co}_3\text{O}_4/\text{PMS}$ system at pH 9.0, so that the reaction pathway, structure-activity relationship, reaction mechanism, and reaction thermodynamics of the DOTP process can be more convincingly elucidated.

2. The DOTPs, occurred in the surface of catalysts with the direct redox reaction between pollutants and oxidizing agents, was close to the catalyst-mediated electron transfer mechanism. The differences need more discussions.

The direct oxidation (DO) mechanism in the DOTPs has substantial differences from the previously reported mediated electron transfer mechanism. As shown in **Figure R5**, according to previous reports, the mediated electron transfer mechanism may involve two different scenarios, namely, electrons are transferred first from the pollutant to the catalyst, and then from the catalyst further to PMS/PDS (Type 1); electrons are transferred from the pollutant to the activated PMS/PDS-catalyst complex (Type 2) (Yun et al., *Environ. Sci. Technol.*, 2018, 52, 7032-7042 and Ren et al., *Environ. Sci. Technol.*, 2019, 53, 14595-14603). By contrast, our DO mechanism is that the catalyst surface first activates the pollutant and the oxidant (PMS/PDS) respectively, and then a direct 2-electron redox reaction occurs between the activated pollutant and oxidant molecules on the catalyst surface. In addition, the surface-stabilized pollutant intermediates and the pathways of surface coupling and polymerization reaction after the DO reaction (**Figure R6**) are also different from those in the mediated electron transfer mechanism.

Figure R5. Substantial differences between DO mechanism and the previously reported mediated electron transfer mechanism.

Figure R6. Schematic diagram of the overall DOTP process.

3. For Figure S35, further explanation was needed for that the reaction process was promoted with the addition of methanol and tert-butyl alcohol.

The dose of methanol and tert-butyl alcohol had no inhibitory effect on the $\text{Co}_3\text{O}_4/\text{PMS}/\text{PhOH}$ system, which supports our conclusion that no free radicals are involved in the DOTP. As for the promoted reaction rate by addition of methanol and tert-butyl alcohol, this may be due to an activation effect of these alcohols on the catalyst surface (Adams et al., Science, 2021, 371, 626-632), which deserves further investigations in future works.

4. For ^{14}C -PhOH isotope labeling experiment, all the labeled carbon remained in the suspension after the reaction. The state of the ~20% removed COD need to be clarified.

Combined with the ^{14}C -PhOH isotope labeling experiment (**Figure 1d**) and the carbon balance calculations via the TGA analysis of the catalyst before and after the reaction (**Figure 2b and Lines 794-797**), we demonstrated that the pollutant (PhOH) was ~100% enriched onto the catalyst (Co_3O_4) surface without mineralization. In this process, the position of TOC changed from the solution to the catalyst surface, but its total amount in the whole system did not change. For COD, however, its definition is different from TOC. COD is an indicative measurement of the degree to which the pollutant can be oxidized. Since direct oxidation (DO) reaction occurred in the overall DOTP process, the COD would decrease to a certain extent without altering the total amount of TOC. Based on the dosage of PMS, we estimate that ~20% of the total electrons embodied in the pollutant was ultimately transferred to PMS, which can well explain the ~20% COD removal in our experiments. This clarification has been displayed in the manuscript (**Lines 117-119, 153-155, and 794-797**).

5. The low cost of DOTPs was showing the potential on sustainable water purification. Have the calculation cover the cost of regeneration and waste disposal?

In the cost evaluation of DOTP compared with adsorption and AOP, we chose granular activated carbon as the catalyst of DOTP. Even under the conditions of one-time use (i.e., replacing the used catalyst with new one), the calculated cost of DOTP was lower than that of adsorption or AOP (**Figure 7c-7e and Supplementary Tables 3-5**). Therefore, even without considering the catalyst regeneration, the DOTP is also cost-advantageous. If we consider the post-treatment of the catalyst after saturation (i.e., the one-time-used granular activated carbon), we may burn it as fuel to further increase the economic value.

REVIEWERS' COMMENTS

Reviewer #1 (Remarks to the Author):

The issues that I raised in the previous review report have been addressed properly. Accordingly, I would like to recommend this work to be considered for possible publication in your journal.

Reviewer #2 (Remarks to the Author):

The authors have well addressed my comments. It now can be published.

Reviewer #3 (Remarks to the Author):

I have read through the revised manuscript with the title "Simultaneous Nanocatalytic Surface Activation of Pollutants and Oxidants for Highly Efficient Water Decontamination". A detailed response to the comments and concerns from the editor and reviewers have been made by the authors, and correspondingly, a comprehensive revision has been made in the revised version of the manuscript. After the revision, the quality of the manuscript has been substantially improved. Therefore, I recommend its publication after minor revisions.

I noticed that the almost no adsorption occurred in the Co₃O₄/PMS/PhOH system, which maybe not support the surface activation of pollutants well as shown in the schematic illustration (Figure 6). In addition, the used catalysts should be disposed as hazardous wastes instead of simply burning. The disposal costs should be covered in the evaluation of DOTP in practical applications.

Response to Reviewer #1's comments

The issues that I raised in the previous review report have been addressed properly. Accordingly, I would like to recommend this work to be considered for possible publication in your journal.

We thank the reviewer for the time and efforts on our work.

Response to Reviewer #2's comments

The authors have well addressed my comments. It now can be published.

We thank the reviewer for the time and efforts on our work.

Response to Reviewer #3's comments

I have read through the revised manuscript with the title “Simultaneous Nanocatalytic Surface Activation of Pollutants and Oxidants for Highly Efficient Water Decontamination”. A detailed response to the comments and concerns from the editor and reviewers have been made by the authors, and correspondingly, a comprehensive revision has been made in the revised version of the manuscript. After the revision, the quality of the manuscript has been substantially improved. Therefore, I recommend its publication after minor revisions.

We thank the reviewer for the time and efforts on our work.

I noticed that the almost no adsorption occurred in the $\text{Co}_3\text{O}_4/\text{PMS}/\text{PhOH}$ system, which maybe not support the surface activation of pollutants well as shown in the schematic illustration (Figure 6).

There is no direct relationship between the adsorption amount and the surface activation. The amount of adsorption is closely related to the specific surface area. Metallic oxides generally exhibit very small amounts of PhOH adsorption due to their small specific surface area. However, the surface activation is the fusion and rearrangement of electron clouds between the organics and the catalyst surface. In the $\text{Co}_3\text{O}_4/\text{PMS}/\text{PhOH}$ system, the small adsorption (or binding) of PhOH on Co_3O_4 surface is able to achieve the surface activation in the reaction process. This is supported by a recent study about the self-catalytic system for oxidative removal of pollutants (Huang et al., Environ. Sci. Technol. 2021, 55, 15361-15370), in which almost no adsorption occurred, but pollutants could be activated on the catalyst surface.

In addition, the used catalysts should be disposed as hazardous wastes instead of simply burning. The disposal costs should be covered in the evaluation of DOTP in practical applications.

We highly appreciate the reviewer's suggestion. In this work, we evaluated the main cost of DOTP and compared it with that of adsorption and AOP technologies. If the used catalysts of DOTP were treated as hazardous wastes, similar consideration should be taken into the adsorption (i.e., regeneration of the used activated carbon) and AOP (i.e., disposal of the generated iron sludge in Fenton system). The specific application costs in practical processes, including the post-treatment of the used catalysts, would be comprehensively estimated on a case-by-case basis, which warrants further investigation in the future.